# Evaluating Student Knowledge Assessment Using Machine Learning Techniques

**Nuha Alruwais [1] and Mohammed Zakariah [2,***

1   Department of Computer Science and Engineering, College of Applied Studies and Community Services, King Saud University, P.O. Box 22459, Riyadh 11495, Saudi Arabia
2   Department of Computer Science, College of Computer Science and Information, King Saud University, P.O. Box 11442, Riyadh 11574, Saudi Arabia
*   Correspondence: mzakariah@ksu.edu.sa

**Abstract:** The process of learning about a student's knowledge and comprehension of a particular subject is referred to as student knowledge assessment. It helps to identify areas where students need additional support or challenge and can be used to evaluate the effectiveness of instruction, make important decisions such as on student placement and curriculum development, and monitor the quality of education. Evaluating student knowledge assessment is essential to measuring student progress, informing instruction, and providing feedback to improve student performance and enhance the overall teaching and learning experience. This research paper is designed to create a machine learning (ML)-based system that assesses student performance and knowledge throughout the course of their studies and pinpoints the key variables that have the most significant effects on that performance and expertise. Additionally, it describes the impact of running models with data that only contains key features on their performance. To classify the students, the paper employs seven different classifiers, including support vector machines (SVM), logistic regression (LR), random forest (RF), decision tree (DT), gradient boosting machine (GBM), Gaussian Naive Bayes (GNB), and multi-layer perceptron (MLP). This paper carries out two experiments to see how best to replicate the automatic classification of student knowledge. In the first experiment, the dataset (Dataset 1) was used in its original state, including all five properties listed in the dataset, to evaluate the performance indicators. In the second experiment, the least correlated variable was removed from the dataset to create a smaller dataset (Dataset 2), and the same set of performance indicators was evaluated. Then, the performance indicators using Dataset 1 and Dataset 2 were compared. The GBM exhibited the highest prediction accuracy of 98%, according to Dataset 1. In terms of prediction error, the GBM also performed well. The accuracy of optimistic forecasts on student performance, denoted as the performance indicator 'precision', was highest in GBM at 99%, while DT, RF, and SVM were 98% accurate in their optimistic forecasts for Dataset 1. The second experiment's findings demonstrated that practically no classifiers showed appreciable improvements in prediction accuracy with a reduced feature set in Dataset 2. It showed that the time required for related learning objects and the knowledge level corresponding to a goal learning object have less impact.

**Keywords:** student knowledge assessment; machine learning; gradient boosting machine; logistic regression; predictive features; performance prediction

## 1. Introduction

High-quality education requires both the education system and students to meet high standards. Philosophers often offer guidelines and benchmarks for enhancing performance and evaluating student knowledge assessments to meet these standards, but the current system still has deficiencies that need to be addressed [1]. In addition to this, the systems still need to catch up. Therefore, researchers concluded that technology might be a significant component in analyzing the defects in the current system and why it lags.

Additionally, technology facilitates decision-making by producing reports and graphs for analytical purposes [2]. In this context, ML emerges as a cutting-edge methodology with several applications that can anticipate outcomes from data [3]. The goal of ML approaches or educational data mining (EDM) is to model [4], find important hidden patterns, and extract information that can be applied in educational situations [5]. Additionally, ML techniques are used in the academic sector [6] to represent a variety of student attributes as data points in big databases. By fulfilling various objectives, such as extracting patterns, anticipating behavior, or spotting trends [3], these strategies can be helpful in a variety of sectors and can help educators deliver the most effective teaching methodologies and track and monitor their students' development [7].

The primary goal of ML is to foresee future scenarios or events that are unknown to computers [8–11]. Data mining (DM) and ML enable data processing, patterns, learning models, analyzing, scheduling, problem solving, predicting, and object manipulation. One of the critical benefits of ML is that it can finish difficult and time-consuming jobs, freeing up time that can be used for other purposes. In educational institutions, ML has been applied in a variety of ways, including automating administrative and procedural chores, developing curricula and content, teaching, and student learning processes [10,12].

ML techniques are currently very sophisticated and are capable of conducting more than just grading examinations using the answer key. In addition to conducting more conceptual assessments like scoring essays [13] or student engagements [14,15], they can provide data about students' knowledge and performance. ML techniques are able to observe student behavior and assess how well they performed. This ability has increased due to ML, enabling decision-makers to retrieve information from data for judgments and policies. Instructors and institutions can study the educational database using powerful techniques like ML and DM.

Moreover, ML's application in an educational database is designated as EDM [9]. How seriously a student takes achieving educational goals is demonstrated by the assessment and appraisal of that student's performance, which also offers details on how well a student learns, how motivated a student is to study, or how successful the teaching technique was [16]. The evaluation's outcomes help teachers decide what is best for the student's growth and how to give valuable feedback. The study considered all of these variables by choosing the proper traits, because each student's unique characteristics, such as personality, motivation, self-efficacy, intelligence quotient (IQ), and self-control, are closely related to their success [17].

Further, anticipating and analyzing student performance [18] are essential for helping teachers identify students' areas of weakness, while supporting them in raising their grades. Students can do the same as when managers enhance their processes [19]. Teachers can identify students who are doing poorly and intervene early in the learning process to implement the appropriate interventions, thanks to the timely prediction of student performance [20].

To produce a more precise estimate of the response variable, the learning process for GBM sequentially fits new models. Powerful ML algorithms, such as GBM, have demonstrated significant success in various real-world applications [21]. They can be learned concerning different loss functions, for example, and are highly adaptable to the application's specific needs. The fundamental concept of the GBM technique is to build the new base learners to have a maximum correlation with the ensemble's overall negative gradient of the loss function. The learning process will produce consecutive error fitting if the error function is the traditional squared error loss. However, the loss functions employed can be random in order to provide more excellent intuition. Generally, it is up to the researcher to decide which loss function to use. A wide range of loss functions have been derived so far, and one has the option of constructing their own task-specific loss.

The primary motivation for our investigation was the need for systematic and thorough studies evaluating the prediction of student academic performance using various ML models. Moreover, the primary goal of this research was to review and examine the

critical predictive variables and ML algorithms employed to forecast students' academic success. Therefore, this effort aims to respond to the following research questions: What are the main predictive factors considered when rating student performance? What are the most critical ML algorithms for predicting student performance? What are the results and accuracy of the ML algorithms?

Figure 1 provides a clear and concise overview of the research approach used in the paper, highlighting the different stages and components of the approach and how they contributed to evaluating student knowledge assessment using ML techniques. Figure 1's knowledge assessment/evaluation survey was used to gather the necessary data from students, which was, subsequently, individually labelled and entered into the existing dataset by the evaluators. Effective models with good performance were discovered after the ML models had been fitted to the data. The essential elements were then chosen from the original information and re-added to the smaller database for use in the models and for student evaluation. The data from the smaller database was also used to retrain the ML models. ML models can now evaluate students with high accuracy, meaning that human assessors are no longer required to predict student achievement.

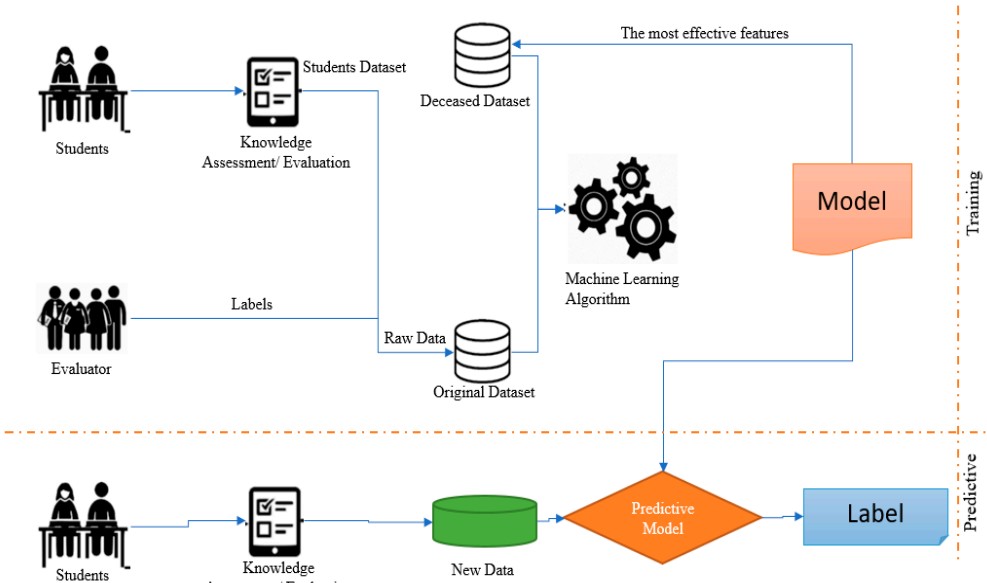

**Figure 1.** Overview of the research approach related to evaluating student knowledge assessment using ML techniques.

The ML technique used to evaluate student performance is shown in Figure 2 below, a dataset where the learning dataset and the independent test set are separated. We used 5-fold cross validation with the training and validation sets from the learning dataset. The best prediction model and dataset are evaluated after the feature selection, classifier construction, and evaluation with the training set.

Our study was driven primarily by the need for more systematic and thorough studies evaluating the prediction of student academic performance and knowledge assessment using various ML models. Consequently, the primary goal of this work was to review and describe the essential predictive variables and the ML algorithms used to forecast students' performance and gauge their knowledge. The study's conclusions provide evidence for mapping and evaluating current knowledge, identifying research gaps, and making recommendations for future studies in this area.

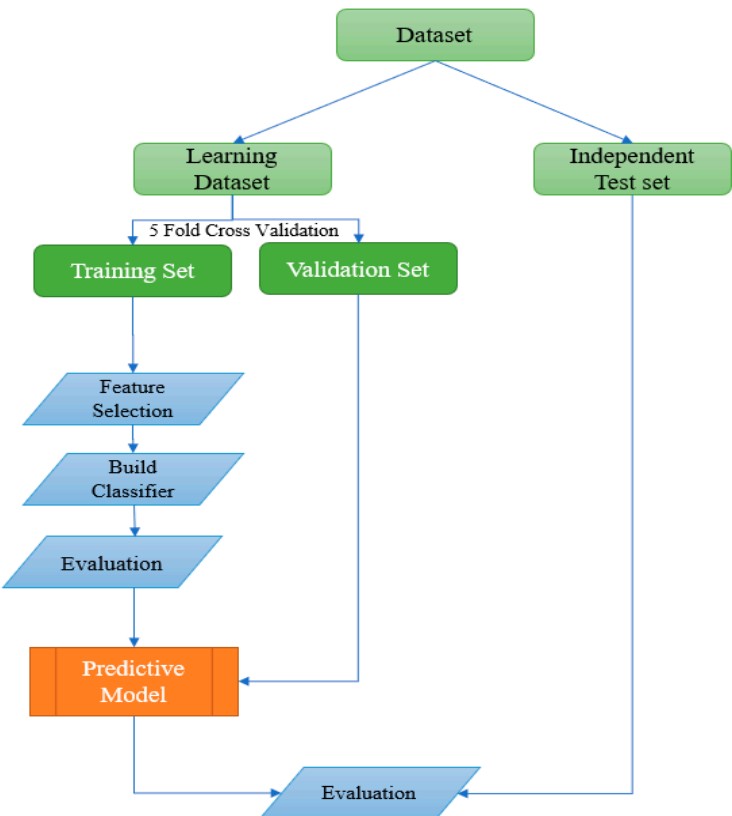

**Figure 2.** Flowchart referring to the ML process used to assess student performance.

The main goal of this research was to create an ML model for categorizing students into various groups according to their knowledge levels. With this goal in mind, the significant contribution of this paper is as follows:

- This paper aimed to identify a group of characteristics or traits that influence students' automatic knowledge classification and, also, to determine the role of ML in conceptualizing and evaluating student education, as well as the challenges and risks that need to be considered;
- This paper used the two experiments to understand the impact of reducing the feature vector on the prediction accuracy of the classifiers and to identify the best technique for simulating students' automatic knowledge classification;
- To identify a group of characteristics or traits that influence students' automatic knowledge classification, this study used seven different classifiers, including SVM, LR, RF, DT, GBM, GNB, and MLP;
- This paper presents the analysis for identifying the minor correlated variables and removing them from the dataset to create a smaller dataset (Dataset 2) providing more focused research and understanding of the impact of a minor feature set on the performance of the classifiers.

The paper is organized as follows: Section 1 discusses the introduction to the work, then Section 2 is dedicated to a state-of-the-art literature review with recent references. Next, Section 3 describes the methodology, including dataset preparation, the proposed process, evaluation, the performance indicators, and experimentation. Then in Section 4, the results are discussed and supported by accurate figures and tables. Finally, Section 5 is the discussion, and Section 6 is the conclusion, followed by relevant references.

## 2. Literature Review

The integration of ML in various facets of education has been the subject of numerous research projects, and multiple techniques and tools have been employed to carry out such tasks. Evaluation of student performance and knowledge assessment is one of these components. There are many different ways to assess student achievement [22–24]. Numerous studies assessed students' performance and assessment in general [25–27], whereas other research assessed students for a particular goal, including academic accomplishment [5]. For each of the activities listed above, some cutting-edge research studies that assessed student performance in various ways are discussed below.

In this work [8], the authors offered various strategies for suggesting online learning systems to improve the construction of the learning management system (LMS) using natural language processing technology. Some of these methods involve content-based filtering, collaborative filtering, utility filtering, knowledge filtering, demographic filtering, community filtering, and hybrid filtering.

In another work [9], researchers reported a preliminary investigation into the development, application, and delivery of the LMS. The paper provides an overview of learning analytics, which integrates data with learning. The study concluded that the most prominent models in the literature are those for learning analytical models. These models involve four steps: gathering pertinent data, reporting it, making predictions, taking action, and modifying the learning environment in response to the data. Unfortunately, the report does not refer to particular ML techniques that could be used in the model.

Similarly, in [11], the authors provided an overview of EDM by going through its core concepts. Both studies included summaries and analyses of the EDM industry and its processes, as well as inclusive learning analytics, but they did not adhere to the requirements for a systematic literature analysis [11–28].

Another work [29], produced another analytical review of the literature to offer an overview of EDM [30]. Some of the strategies used in the study were forecasting, segmentation, outlier detection, process mining, relationship mining, social network analysis, data distillation for human judgment, text mining, knowledge tracing, discovery with models, and non-negative matrix factorization. However, the authors of this study needed to adhere to the standards for a systematic literature review, which typically involves a rigorous and systematic search, appraisal, and synthesis of the literature on a topic. Additionally, the study did not focus specifically on ML methods, a specific field of study within EDM that involves using ML techniques to analyze educational data. Therefore, the analysis needed a comprehensive and in-depth examination of the ML methods for EDM.

Much data was recently acquired on student performance and assessment indicators with actions like reading files, engaging in forums, sending messages, or viewing suggested links, by some studies that were recently applied to an online learning environment. However, when evaluating the research on learning analytics, it was found that many of the previous studies concentrated on forecasting student outcomes [18,31], at-risk students [19], and student performance [25].

In [5], the authors provided an anonymized dataset with 3,568,825 instances utilized to predict dropout likelihood concerning student assessment and performance evaluation. Free/reduced lunch eligibility and student demographics were used as features and were pulled from various school districts, educational institutions, and agencies.

In addition, the classification method for LR was used to define the relationship between a discrete response variable and one or more independent variables. Although LR is utilized for classification problems when the response variable has two or more classes, linear regression is typically employed to predict response variables with continuous values. According to [3], LR has emerged as one of the most popular approaches to categorization issues across a range of industries.

Reflective research on artificial intelligence and machine learning (AIML) by [12] examined the themes and their development, while pointing out the recent rise in interest in profiling and analytics. An overview of the use of deep learning and artificial intelligence

in teaching and learning is given in the paper. However, the study does not emphasize using ML algorithms for online education. A survey-based study by [11] used educational data to create models for enhancing academic performance and institutional effectiveness.

Learning analytics has been used in several types of research to forecast student assessment performance using various ML methods. The majority of experts concur that student participation affects their performance evaluation [3,32,33]. However, only some studies have examined learning analytics to forecast student engagement concerning performance. They must still look at ways to improve and inspire students or assess their knowledge.

The list of the past publications cited and the techniques and findings in assessing student knowledge is provided below in Table 1.

**Table 1.** List of past paper references with the methodology used and the results.

| Ref | Dataset | Methodology | Results |
|-----|---------|-------------|---------|
| [8] | • One hundred and nine publications from all sources and criteria were examined using a systematic literature review (SLR). Of those publications, 55 papers were chosen as study candidates based on their titles and abstracts with the research questions. | • ML based on smart LMS for online learning. | • Revealed collaborative filtering;<br>• They created reliability standard (RS) specific methodologies and instructional strategies for online learning. |
| [10] | N/A | • EDM;<br>• Learning analytics;<br>• ML. | • A significant portion of the algorithms that get better with use was the focus of the ML;<br>• The art of obtaining valuable information from sizable datasets. |
| [11] | • Reviewed 36 out of 420 research publications from 2009 to 2018 and analyzed them using the SLR method. | • EDM methods used;<br>• Interpretation of prediction model;<br>• Optimize learning path or personalized learning resources. | • Provided valuable insights on methods for enhancing pedagogical processes, predicting student performance, comparing the accuracy of DM, and developing open-source tools and algorithms. |
| [34] | • Reviewed 13 out of 199 research publications. | • A comparison of the traditional multilinear regression model's predictive power;<br>• Hybrid analysis, implementations of the extreme GBM stacking ensemble, SVM, RF, and artificial neural networks (ANN). | • The correlation coefficient between the Det and the performance gap between XGBoost and RF was 0.7911;<br>• Solutions proposed to reduce dropout in distance learning. |

**Table 1.** *Cont.*

| Ref | Dataset | Methodology | Results |
|-----|---------|-------------|---------|
| [35] | • Based on various aspects of student interactions (social presence, cognitive presence, and teaching presence) in the virtual environment, the data collected from Moodle was used to construct 13 different datasets. | • Receiver operating characteristics–area under ROC curve (ROC-AUC);<br>• ML techniques. | • The models' performance varied from semester to semester, but the top ones could spot at-risk students in the first week of the class with an AUC-ROC of between 0.6 and 0.9. |
| [36] | • A total of 65 features were identified as suitable features;<br>• To optimize data collection, the authors reduced the number of characteristics to 35. | • The system used RF, SVM, LR, and ANN algorithms as classifiers;<br>• The Boruta algorithm was employed. | • The results showed that ML models help evaluate students;<br>• Positive traits were identified to promote learning;<br>• SVM performance was enhanced, and its accuracy score of 0.78 was the highest of all the models. |
| [28] | Data was collected from the Moodle logs of introductory programming courses from the information and communication technologies (ICT) undergraduate program at the Federal University of Santa Catarina (UFSC). | • DM and ML;<br>• GNB classifiers. | • The results showed that ML models help evaluate students. |

## 3. Materials and Methods

This section delves into the details of the methodology put forth in our proposal. It includes a comprehensive discussion of the steps involved and their reasoning. The aim is to provide a clear understanding of the approach taken in this study and demonstrate the efficacy of the method used. Pre-processing was performed to prepare the dataset for this study, a correlation analysis of the attributes was conducted, and the dataset was split into various classes and attributes. Furthermore, this section delves into the classification of student performance using seven different classifiers and examines their performance through multiple indicators and experimentation on the proposed methodology.

### 3.1. Dataset

In this study, ML classification models were designed and assessed using a dataset obtained from an e-learning environment. The dataset was prepared through a six-step process that monitored and evaluated the students' progress and understanding at each stage. The six-step process included defining the attributes, gathering the data, cleaning and pre-processing, organizing the data, classifying the data, and correlation analysis of the attributes. The resulting dataset reflects five student characteristics observed while interacting with the learning platform. These five attributes in the dataset were broadly categorized into three fundamental types of attributes: individual behavioral attributes, attributes related to exam scores, and the objective attribute of knowledge level, as seen in Table 2.

**Table 2.** Data description.

| Attribute Abbreviation | Attribute Name | Attribute Type | Attribute Description |
|:---:|:---:|:---:|:---:|
| STG | Degree of study time for the goal object | Individual behavioral | The measure of time taken by students for target learning objects |
| SCG | Degree of study counts for the goal object | Individual behavioral | The measure of target learning object repetition |
| STR | Degree of study time for the related object | Individual behavioral | The measure of study time of the student for objects related to the target object |
| LPR | Learning percentage for the related objects | Exam score related | The score of the user for objects related to the target objects |
| PEG | Performance in exams for goal object | Exam score related | The score of the user for the target objects |
| UNS | User knowledge state | Knowledge level | The knowledge level of the user |

This study employed the k-nearest neighbor (k-NN) approach to classify the user knowledge classes. The k-NN approach is a widely used ML algorithm for classification and is known for its simplicity and effectiveness. The aim was to categorize the user knowledge into different classes accurately, and the k-NN approach proved to be an effective method for achieving this goal. Four classes of students were retrieved: very low, low, middle, and high. The distribution of the students over the classes in the dataset is shown in Figure 3 below. As seen, the allocation was made according to the four student classes. Around 30% of the students were in the class with high-performing students, 28% in the class with average-performing students, and 30% and 12% were in classes with low and very low-performing students, respectively.

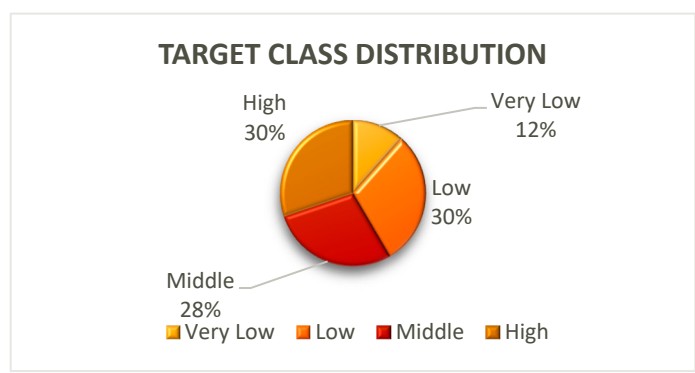

**Figure 3.** Distribution of the four classes of students.

There were 403 instances, or records, in the dataset, each with values for the different attributes. The attributes and their descriptions are presented in Table 2. They are STG, SCG, STR, LPR, PEG, and UNS, respectively. Study time for the goal/target study object and comparable study objects is measured by the acronyms STG and STR, respectively. The exam scores students received for study materials related to the target objects and specifically for goal objects were LPR and PEG, respectively. The number of times a student repeated a single target study item to understand it was referred to as SCG by [37], and UNS is the property's name that denotes the user's knowledge level.

The retrieved features were combined, and the combined feature vector was then optimized using a method based on the Pearson correlation coefficient to choose the most compelling features, while eliminating the unnecessary ones.

This study used the five attributes PEG, LPR, STR, SCG, and STG as dependent variables and UNS as the independent variable to categorize the students depending on

their knowledge. A Pearson correlation analysis was performed to determine the link between the dependent and independent variables [38]. The five qualities, PEG, LPR, STR, SCG, and STG, are all positively linked with the variable UNS in the sample, according to Figure 4. The analysis only revealed a negative association between the variables (PEG, LPR, STG, STR), the student's objective learning object, and the score and study time for other learning objects. The students' knowledge assessments correlated strongly with the target object's score. The dataset with five attributes, including PEG, LPR, STR, SCG, and STG, was Dataset 1. On the other hand, the UNS, the student's degree of knowledge, and the STR, or the amount of time the student spends on related learning materials, had the slightest correlation. As a result, the feature vector was simplified to be composed of the features including the PEG, LPR, SCG, and STG. The least correlated variable, 'STR', was removed from the dataset to create a smaller dataset, termed Dataset 2.

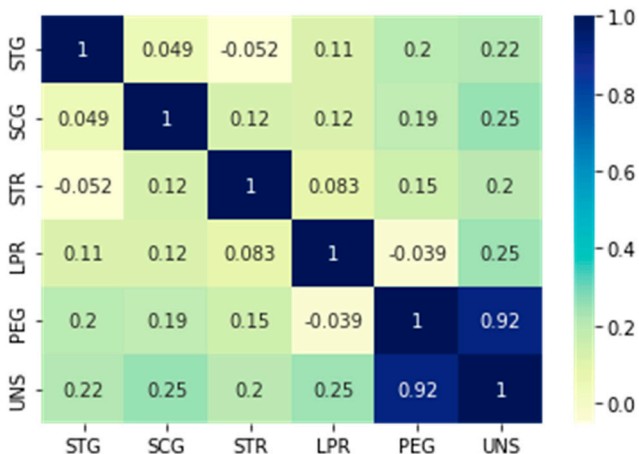

**Figure 4.** Pearson correlation analysis among the features of the dataset.

### 3.2. Proposed Methodology

This paper suggested a classification approach for student's knowledge according to their level of expertise, depicted in Figure 5. The first set consisted of data gathering and analysis. After studying the correlation between the features, the dataset was split into test and training datasets. Then, ML models were created using training data to categorize the pupils based on their knowledge levels. Next, different assessment techniques were used to assess the categorization models. The later parts provided more details regarding the suggested methodology's various stages. Finally, the features were chosen based on correlation analysis, and the same process was applied to the smaller dataset, to identify the ideal set of qualities that contribute to the automatic knowledge classification of students.

### 3.3. Data Classifications and Evaluation Methods

The e-learning dataset was used in this study to highlight some performance aspects against classification algorithms. The classifiers used to create the knowledge evaluation model were RF, SVM, LR, DT, GBM, GNB, and MLP.

The evaluation of the classification algorithms was the primary focus of this research project. The classification accuracy, precision, recall, error rates in prediction, and the area under the receiver operating characteristic curve were the metrics employed in this study to assess the classification algorithms. The various classifiers and assessment measures used in this study are discussed in more detail in this section.

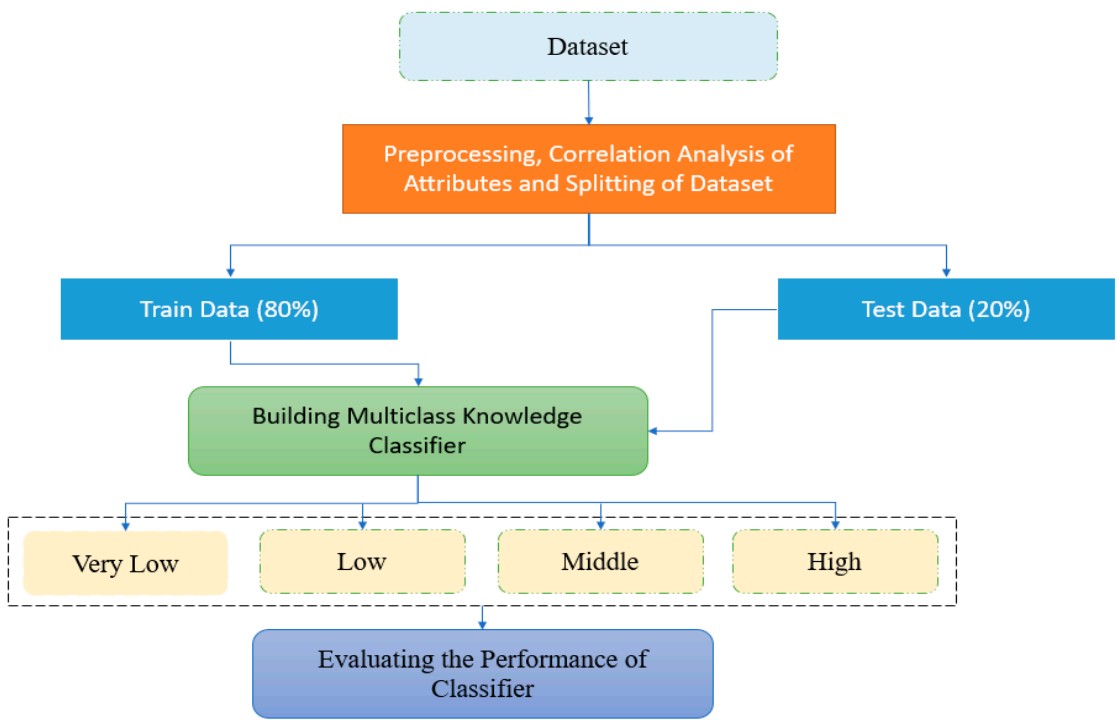

**Figure 5.** Proposed methodology.

### 3.4. Classifiers

3.4.1. Random Forest Classifier (RF)

RF is an ensemble learning method combining multiple decision trees to create a more robust model. This approach is a forest or a collection of trees, as the name would imply. The tree-based classifier makes trees in stages and selects the best tree using a voting method. It also provides a relatively reliable indication of the usefulness of the feature. This process has four steps: 1. Choose a group of samples at random; 2. Create trees and obtain the forecast outcome; 3. Assign each tree a voting score; 4. Choose the tree with the most votes. In RF, each decision tree makes a prediction based on the features of a student, and the final prediction is made based on the majority vote of the decision trees. It allows the algorithm to handle non-linear relationships between features and target classes, making it a popular choice for many classification problems.

A classifier and a regressor can be created using the supervised learning technique RF [39]. The RF classifier was a viable option for the smaller dataset because it does not show overfitting, which is typical when working with a smaller dataset.

$$\text{Regression Equation}: \int_{rf}^{B}(x) = \frac{1}{B}\sum_{b=1}^{B} T_b(x) \tag{1}$$

Here, Equation (1) represents the regression equation for the RF classifier. Where $B$ is the number of data points, $rf$ is the value returned by the model, and $b$ is the actual value of the data point $i$.

3.4.2. Support Vector Machines (SVM)

SVM is a popular ML algorithm that can be used to solve multiclass classification problems, including evaluating student performance. SVM works by finding the hyperplane that separates the data into different classes with the maximum margin. The data points closest to the hyperplane are called support vectors, which play a crucial role in determining the hyperplane. In the context of student performance evaluation, SVM was used to predict a student's class based on their attributes, such as grades and attendance.

The algorithm can handle multiclass classification problems by creating multiple binary classifiers and combining their results to arrive at the final prediction [40]. The results were then contrasted as one against the rest and one against another. The classifier with the highest accuracy score is considered the final result. The real benefit of SVM is that, with no adjustments, it also performs well for data that can be separated linearly.

The following is the equation related to the SVM:

$$y_i(w. \, x_i + b) \geq 1 - \zeta_i, \, i = 1 \ldots \ldots m \tag{2}$$

The above equation, Equation (2) represents the decision boundary of an SVM model. In SVM, the goal is to find the optimal hyperplane that separates the data points into different classes by maximizing the margin between the classes. Where $w$ is the weight vector, $x$ is the input vector, and $b$ is biased.

The SVM optimization problem is given by:

$$f\,(w) = \frac{1}{2}\,\|w\|^2, \, g(w, \, b) = y_i(w. \, x_i + b) - 1, \, i = 1 \ldots m \tag{3}$$

As above, Equation (3) shows the SVM optimization problem or Lagrangian dual problem: instead of minimizing over $w$, $b$, subject to constraints involving $a$'s, we can maximize over $a$ (the dual variable) subject to the relations obtained previously for $w$ and $b$.

### 3.4.3. Logistic Regression (LR)

LR is a statistical method used for solving classification problems. It is a generalized linear model used to model the relationship between a binary outcome variable and a set of input features. The basic principle of LR is to find the best linear combination of the input features that maximize the likelihood of the observed outcome.

In evaluating student knowledge assessment, LR can be used to predict students' proficiency levels in a particular learning object. It can also be used to predict the student's performance in future assessments, improving the teaching and learning process. LR is a simple, interpretable, and efficient model that can handle linear and non-linear relationships between student performance and knowledge and the characteristics that influence them. Additionally, LR can be used for multiclass problems in which the dataset has more than two classes. It uses a logistic function to model the relationship between the independent and binary dependent variables. In the context of student performance evaluation, the independent variables could be student attributes, such as study habits, socio-economic status, and others. In contrast, the dependent variable is the student's performance level, divided into multiple classes. LR estimates the probabilities of the response variable being in each category and assigns the class with the highest probability as the predicted class.

LR [41] calculates an event's probability (0 and 1) given a collection of independent factors. The overfitting of LR models is relatively standard. Therefore, standardizing data is necessary before processing it.

The logistic function used in LR models is the softmax function, which converts the linear combination of the independent variables into a probability distribution over the different categories. The softmax function is defined in Equation (4):

$$P_i = \frac{e^{z_i}}{\sum e^{z_j}}, \, for \, i = 1, \, 2, \, \ldots, \, k \tag{4}$$

where $P_i$ is the predicted probability of category $i$, $z_i$ is the linear combination of the independent variables for category $i$, and $k$ is the total number of categories. To compute $z_i$ for each category $i$, the LR model estimates the coefficients of the independent variables in a similar way to the binary case. However, instead of estimating a single set of coefficients, LR for multiple categories estimates a separate set of coefficients for each category, resulting in a matrix of coefficients.

LR for multiple categories, models the probability of each category given a set of independent variables using the softmax function. The model estimates a matrix of coefficients to compute the linear combination of the independent variables for each category, and standardizing the data is necessary to avoid overfitting.

### 3.4.4. Decision Tree Classifier (DT)

The DT classifier is an algorithm for supervised learning tasks, such as classification and regression. It works by recursively partitioning the dataset into subsets based on the values of the input features. The goal is to create partitions that will result in the most significant separation of the classes. The partitioning process results in a tree-like structure, with each internal node representing a feature test and each leaf node representing a class label. The basic principle of a DT is that it learns to approximate any complex function by training on a set of input–output pairs. The training process is based on finding the best feature test at each internal node, resulting in the most significant separation of the classes. The final decision tree is a set of feature tests, with each test representing a path from the root of the tree to a leaf node.

In evaluating the student knowledge assessment, a DT classifier can classify students into different proficiency levels for a particular learning object, such as novice, beginner, intermediate, or expert. A DT can also be used to predict the student's performance in future assessments, which can be used to improve the teaching and learning process. A DT can learn non-linear relationships between the student's performance and knowledge, and the characteristics that influence them, and it is simple to interpret and explain the results. Additionally, a DT can classify multiclass problems in which the dataset has more than two classes.

Using two different types of nodes for the trees, decision nodes and leaf nodes, a decision tree classifier [42] is a tree-based classifier. Decision nodes produce decisions and have many branches, whereas leaf nodes indicate the outcomes of the decisions and do not have any additional components. The decision-making process is based on the dataset's features. An attribute selection measure is used to select the optimal feature for both the root node and sub-nodes. A question is posed in a decision tree, and the answer (yes/no) determines which subtrees are included. Here, the starting decision node represents the root node, which was followed up by the subtree, as shown in Figure 6.

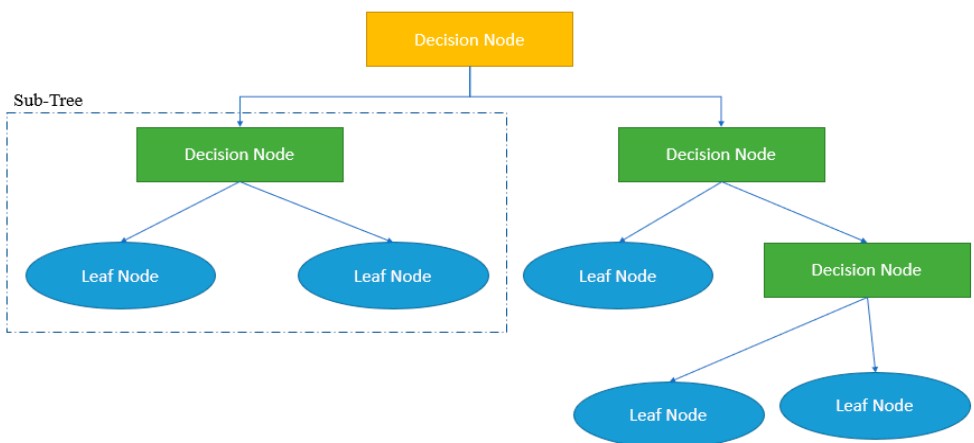

**Figure 6.** Decision tree algorithm in ML.

### 3.4.5. Gradient Boosting Machine (GBM)

The GBM [43] is an ML algorithm for supervised learning tasks, such as classification and regression. It is an ensemble learning method that combines the predictions of multiple base models to improve the overall performance. The GBM works by iteratively adding new base models, such as decision trees, to the ensemble, where each new model is trained to correct the errors made by the previous models. The algorithm uses gradient descent

to minimize the errors made by the ensemble. In each iteration, the algorithm fits a new model to the negative gradient of the loss function, which measures the error made by the ensemble. The final ensemble is a weighted sum of the base models, where the training process determines the weights.

In evaluating the student knowledge assessment, the GBM can classify students into different proficiency levels for a particular learning object, such as novice, beginner, intermediate, or expert. By using a GBM, the system can automatically learn the relationship between the student's performance and knowledge and the characteristics that influence them. A GBM can also predict the student's performance in future assessments, which can be used to improve the teaching and learning process.

A GBM is a greedy classification technique based on trees. To reduce the inaccuracy of the previous model, fresh decision trees and weak classifiers are first built. Then, finding split spots to split the tree using the greedy algorithm is the most effective way to minimize the objective function.

### 3.4.6. Gaussian NB Classifier (GNB)

The NB classifier family, which includes the GNB [35], uses the Bayes theorem to classify data. The GNB assumes that the continuous values associated with each feature are dispersed according to Gaussian distribution. When looking for features affecting the classification problem, naive Bayes classifiers can find them using incredibly small quantities of data. A GNB is an ML algorithm that can be used for multiclass problems in student performance evaluation. It is based on the Bayes theorem and assumes that the features are conditionally independent given the class variable. The algorithm calculates the probability of each class given the features, and the class with the highest probability is assigned as the final prediction. A GNB is simple to implement and can handle both continuous and discrete features. However, it may perform poorly when the features are highly correlated.

A GNB supports continuous-valued features in Equation (5), which also models each according to Gaussian (normal) distribution.

$$P(x_i|y) = \frac{1}{\sqrt{2\pi\sigma^2 y}} \exp\left(-\frac{(x_i - \mu_y).^2}{2\sigma^2 y}\right) \tag{5}$$

where $P(t) = \pi$. Parameters $\pi$t and $\mu$ can be learned using the maximum likelihood. The above equation uses the Gaussian distribution and the dependence relation of $x_i$ which is encoded in the covariance matrix.

### 3.4.7. Multi-Layer Perceptron (MLP)

Another ANN technique with several layers is the MLP. The MLP is a type of ANN used for supervised learning tasks, such as classification and regression. It comprises an input layer, one or more hidden layers, and an output layer. Each layer is made up of a set of neurons, which are connected to the neurons in the previous and subsequent layers via a set of weights. The layers within it are dense and interconnected, allowing it to transform any input dimension into the required dimension [44]. The MLP is a term describing a neural network with several layers. We combine neurons in a neural network, so that some of their outputs also function as inputs.

Linear issues can be solved in a single perceptron, but non-linear examples must be better suited. An MLP can be used to resolve these challenging issues. The network weights are set in a random order before starting the training. After completing the learning step by using the training data ($x_1$, $x_2$, $y$), the model is validated. In the training set, data $x_1$ and $x_2$ are the input, and $y$ is the corresponding expected output of the input data.

The basic principle of the MLP is that it learns to approximate any complex function by training on a set of input–output pairs. The training process is based on adjusting the weights of the connections between the neurons to minimize the error between the

predicted output and the true output. In evaluating the student knowledge assessment, an MLP can classify students into different proficiency levels for a particular learning object, such as novice, beginner, intermediate, or expert. An MLP can also be used to predict the student's performance in future assessments, which can be used to improve the teaching and learning process. An MLP can learn non-linear relationships between the student's performance and knowledge and the characteristics that influence them. Additionally, an MLP can be used to classify multiclass problems in which the dataset has more than two classes.

### 3.5. Evaluation Metrics

Various evaluation metrics are used to determine the classifier's confusion or prediction error [45]. A special kind of contingency table with two dimensions, actual and anticipated results, and identical sets of "classes" is used to depict the confusion, as shown in Table 3.

**Table 3.** Confusion matrix.

| | | Predicted | |
| --- | --- | --- | --- |
| | | **Positive** | **Negative** |
| Actual | Positive | True positive (TP) | False negative (FN) |
| | Negative | False positive (FP) | True negative (TN) |

Performance indicators that help comprehend each classifier's performance can be developed using the confusion matrix [33]. The following performance measures are reported utilizing the classifier outputs. The metric used to determine the accuracy of accurate forecasts is called precision (Equation (7)). The ratio of accurately anticipated outcomes to all estimates is known as recall (Equation (8)). When false negatives outweigh false positives, recall is a valuable metric. The ratio of all correct forecasts to all correct and incorrect predictions is called accuracy (Equation (6))

$$Accuracy = \frac{TP + TN}{TP + TN + FP + FN} \tag{6}$$

$$Precision = \frac{TP}{(TP + FP)} \tag{7}$$

$$Recall = \frac{TP}{(TP + FN)} \tag{8}$$

The root mean squared error (RMSE) (Equation (9)), the mean absolute error (MAE) (Equation (10)), the mean squared error (MSE) (Equation (11)), and the root absolute error (RAE) (Equation (12)) are used to evaluate the error measures in the prediction [46].

$$\text{RMSE} = \sqrt{\frac{1}{m} \sum_{i=1}^{m} (py_i - my_i)^2} \tag{9}$$

$$\text{MAE} = \frac{1}{m} \sum_{i=1}^{m} |py_i - my_i| \tag{10}$$

$$\text{MSE} = \frac{1}{m} \sum_{i=1}^{m} (py_i - my_i)^2 \tag{11}$$

$$\text{RAE} = \sum_{i=1}^{m} \frac{|py_i - my_i|}{py_i} \tag{12}$$

where the test size is m, $py_i$ is the predicted value, $my_i$ is the mean of the actual values, and;

The AUC, in addition to these measurements, is computed. Finally, the ratio of correctly categorized positive samples (TPR) vs. wrongly classified negative samples (FPR) across all potential thresholds is plotted on a graph called the receiver operator characteristic (ROC) curve.

### 3.6. Experimentations

The experimentation consisted of two experiments. First, the dataset (Dataset 1) was used for the initial tests in its original state, including all five properties listed in the Data Description section. Then, after the correlation analysis, the least correlated variable was removed from the dataset to create a smaller dataset (Dataset 2), and the same experiment was repeated in the subsequent phase. The following subsection presents and discusses the findings.

In this study, the various categorization models were constructed and assessed using factual data from an e-learning system. There were 403 instances of actual student data in the collection. Moreover, 80% of the dataset was used for training, and 20% was used for testing, as part of the classification process. All tests were carried out using Python 3.7.15 and Google Collaboratory, a cloud-hosted version of the Jupyter Notebook. The knowledge assessment methodology divided pupils into groups: very low, low, middle, and high. The seven classifiers were utilized in this experiment, as was covered in the previous section. Here, classifiers based on neural networks, regressors, and trees were used. Table 4 displays each classifier's hyperparameters used in the current investigation. A hyperparameter is a parameter that is established before the learning process begins. These programmable options can directly influence the effectiveness of a model railway.

**Table 4.** Hyperparameters of each classifier.

| No. | Classifier | Hyperparameters |
| --- | --- | --- |
| 1 | RF | n_estimators = 200, random_state = 0, criterion = "entropy", max_features = "log2" |
| 2 | SVM | kernel = 'rbf' |
| 3 | LR | random_state = 0 |
| 4 | DT | criterion = "entropy", random_state = 0, max_depth = 5, min_samples_leaf = 5 |
| 5 | GBM | learning_rate = 0.09 |
| 6 | GNB | priors = None, var_smoothing = $1 \times 10^{-9}$ |
| 7 | MLP | hidden_layer_sizes = (10,), activation = 'relu', solver = 'adam', alpha = 0.0001, learning_rate = 'adaptive', max_iter = 2500, max_fun = 17,000 |

## 4. Results

### 4.1. First Experiment

The first experiment aimed to determine the best way to model the automatic classification of student knowledge. The data in this input had five attributes: PEG, LPR, STR, SCG, and STG. It was a multiclass problem since the classifier's output can be "very low, low, middle, high," denoting a student's proficiency in a particular learning object as a novice, beginner, intermediate, or expert, respectively. We employed seven distinct classifiers for modeling, as was covered in the technique section. Table 5 compares the performance of the seven different ML classifiers (RF, SVM, LR, DT, GBM, GNB, and MLP) for evaluating student performance. The performance metrics used for evaluation were: accuracy, precision, recall, MAE, MSE, RMSE, RAE, and the AUC. A visual representation of the performance comparison of the various classifiers with Dataset 1 can be found in Figures 7 and 8.

**Table 5.** Comparison of the classifiers (classifier input: Dataset 1 with five attributes).

| No. | Classifier | Accuracy | Precision | Recall | MAE | MSE | RMSE | RAE | AUC |
|---|---|---|---|---|---|---|---|---|---|
| 1 | RF | 0.96 | 0.98 | 0.89 | 0.09 | 0.04 | 0.21 | 0.051 | 0.98 |
| 2 | SVM | 0.94 | 0.98 | 0.90 | 0.05 | 0.05 | 0.2 | 0.046 | 0.92 |
| 3 | LR | 0.90 | 0.87 | 0.85 | 0.09 | 0.09 | 0.3 | 0.12 | 0.95 |
| 4 | DT | 0.96 | 0.98 | 0.89 | 0.03 | 0.03 | 0.1 | 0.051 | 0.93 |
| 5 | GBM | 0.98 | 0.99 | 0.97 | 0.03 | 0.03 | 0.1 | 0.051 | 0.97 |
| 6 | GNB | 0.90 | 0.94 | 0.91 | 0.09 | 0.09 | 0.3 | 0.11 | 0.96 |
| 7 | MLP | 0.96 | 0.95 | 0.90 | 0.09 | 0.09 | 0.3 | 0.095 | 0.98 |

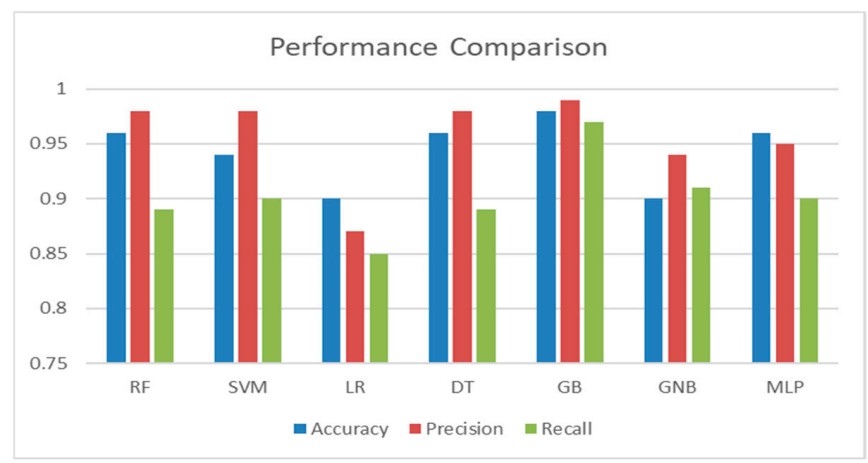

**Figure 7.** Performance comparison of the classifiers (classifier input: Dataset 1 with five attributes).

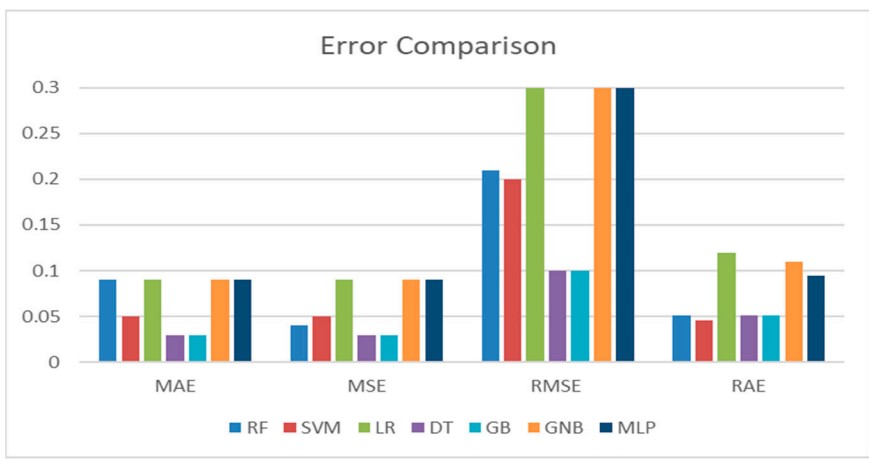

**Figure 8.** Error measure comparison of the classifiers (classifier input: Dataset 1 with five attributes).

According to the data, the GBM exhibited the highest prediction accuracy, 98%. In terms of prediction error, the GBM also performed well. In terms of performance, RF, DT, and MLP were on a par with the GBM. The performance could be better when using linear classifiers like SVM or LR. At this level of testing, the tree-based classifiers performed better.

The tree-based classifiers outperformed the linear classifiers in terms of precision and recall. The GBM made predictions with 99% accuracy, and DT and RF made predictions positively with 98% accuracy. The GBM displayed a 97% recall rate of accurate predictions. With the dataset, the MLP achieved 96% accuracy, 95% precision, and 90% recall, performing well for multiclassification. The GBM outperformed the other classifiers when

the prediction error was taken into account. The error measure comparison of the classifiers is shown in Figure 8.

Regarding the error metrics, the lowest MAE, MSE, and RMSE scores for the DT and GBM classifiers were 0.03, 0.03, and 0.1, respectively, for both DT and GBM. The highest scores were seen in the LR classifier, which were 0.09, 0.09, and 0.3, respectively. The RAE scores ranged from 0.051 (DT and GBM) to 0.12 (LR). The highest AUC score was seen for the GBM classifier (0.97) and the lowest for the SVM classifier (0.92).

### 4.2. Second Experiment

In the second experiment, correlation analysis was used to determine the impact of a more minor characteristic or feature collection. Here, we saw that the UNS, the student's degree of knowledge, and the STR, or the amount of time the student spends on related learning materials, have just the slightest correlation. As a result, the feature vector was simplified to be composed of the following features: PEG, LPR, SCG, and STG. This condensed feature set produced a condensed dataset (Dataset 2). In this smaller dataset, the same methodology was used as in the initial experiment, and the same assessment criteria were used to calculate the results.

Table 6 contains the results from evaluating student performance using the seven different classifiers (RF, SVM, SL, DT, GBM, GNB, and MLP) on a reduced dataset (Dataset 2). With 98% accuracy, 99% precision, and 97% recall, the GBM performed better than all other classifiers, as seen in Table 6. Compared to the other classifiers, the error rates for the GBM were also lower.

**Table 6.** Comparison of the classifiers (classifier input: Dataset 2 with four attributes).

| No. | Classifier | Accuracy | Precision | Recall | MAE | MSE | RMSE | RAE | AUC |
|-----|------------|----------|-----------|--------|------|------|------|------|------|
| 1 | RF | 0.96 | 0.96 | 0.89 | 0.09 | 0.04 | 0.21 | 0.05 | 0.98 |
| 2 | SVM | 0.96 | 0.96 | 0.96 | 0.03 | 0.03 | 0.19 | 0.04 | 0.98 |
| 3 | LR | 0.88 | 0.86 | 0.84 | 0.11 | 0.11 | 0.3 | 0.15 | 0.95 |
| 4 | DT | 0.96 | 0.98 | 0.89 | 0.03 | 0.03 | 0.18 | 0.05 | 0.98 |
| 5 | GBM | 0.98 | 0.99 | 0.97 | 0.03 | 0.03 | 0.19 | 0.05 | 0.98 |
| 6 | GNB | 0.88 | 0.87 | 0.86 | 0.11 | 0.11 | 0.3 | 0.11 | 0.95 |
| 7 | MLP | 0.92 | 0.95 | 0.93 | 0.07 | 0.07 | 0.2 | 0.09 | 0.98 |

According to the table, the highest accuracy and AUC scores were achieved by the GBM and MLP classifiers, with a score of 0.98 for both. In addition, the precision and recall scores for GBM and MLP were also high, with a precision score of 0.99 and 0.95, respectively, and a recall score of 0.97 and 0.93, respectively.

The LR classifier had the lowest accuracy, precision, and recall scores, with an accuracy of 0.88, a precision of 0.86, and a recall of 0.84. The GNB classifier also had a lower accuracy score than the other classifiers, with a score of 0.88. The GBM and MLP were the strongest classifiers regarding accuracy and AUC scores, while the LR and GNB were relatively weaker. A visual representation of the performance comparison of the various classifiers with Dataset 1 can be found in Figures 9 and 10, which demonstrate the peaks for the performance indicators.

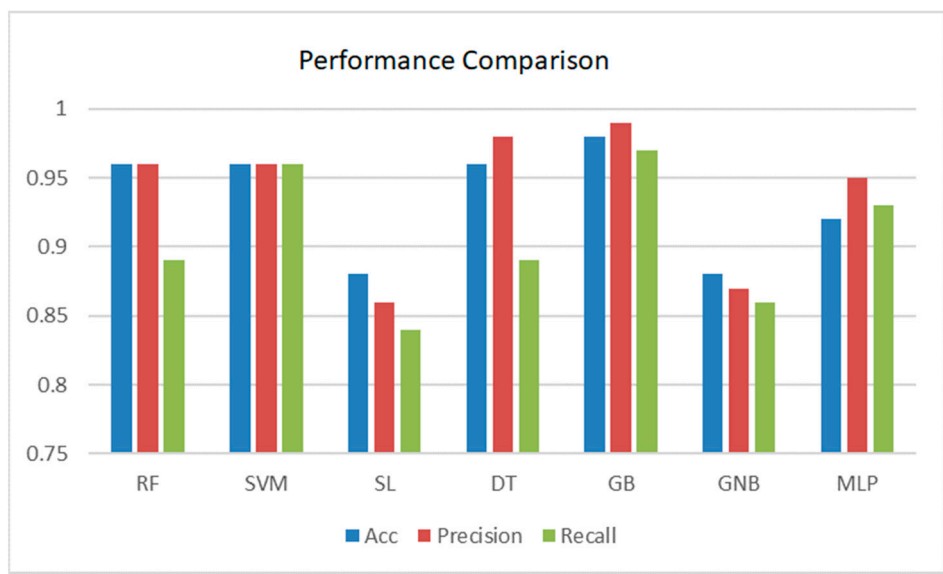

**Figure 9.** Performance comparison of the classifiers (classifier input: Dataset 2 with four attributes).

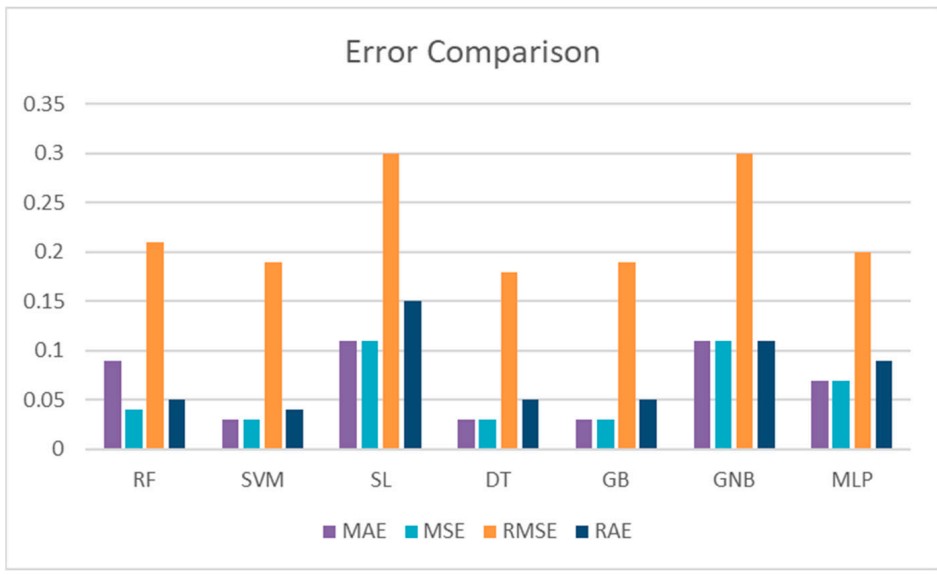

**Figure 10.** Error measure comparison of the classifiers (classifier input: Dataset 2 with four attributes).

### 4.3. Comparative Analysis

Analysis of the data from the first and second experiments revealed exciting findings. Almost no classifiers exhibited any discernible changes in prediction accuracy with a smaller feature set. It showed that the time required for related learning objects and the knowledge level corresponding to a goal learning object have less impact. Therefore, if we do not consider how long a student took to study other learning items, automatic knowledge assessment is still feasible, according to the results. Here, we took into account additional factors like student repetition rates, time, and score related to the goal learning objects. The results revealed a decline in prediction accuracy for the LR and GNB, demonstrating the classifier's dependence on a minor contributing characteristic.

The error metrics, such as MAE, MSE, RMSE, RAE, and AUC, also showed similar patterns with Dataset 1 and Dataset 2, with the GBM classifier consistently performing the best among the seven classifiers. Overall, the GBM classifier performed the best among the seven classifiers in terms of accuracy, precision, recall, and other evaluation metrics.

Figure 11 displays an AUC comparison of the two models using Datasets 1 and 2. In both experiments, we used the one-versus-rest AUC-ROC weighted to prevalence. The AUC results are encouraging in the trial with a smaller feature set. However, even when one attribute is avoided, the prediction systems exhibit exceptional discrimination.

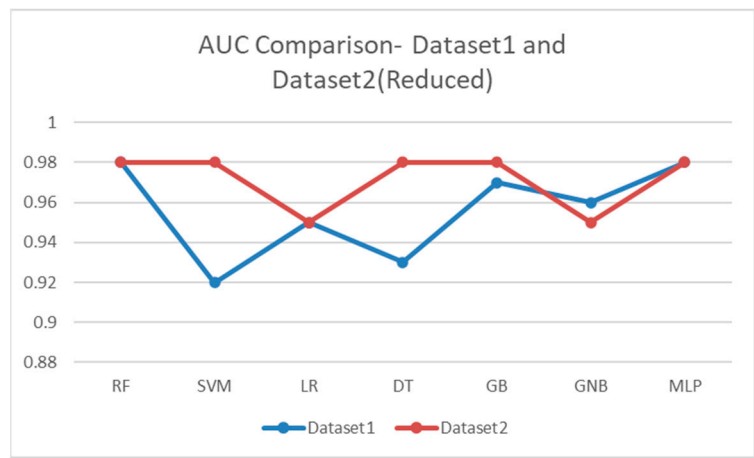

**Figure 11.** AUC comparison between the two datasets (Dataset 1: five attributes, Dataset 2: four attributes).

The confusion matrix was used to assess the effectiveness of a classification model. Table 7 shows the results from evaluating different ML classifiers on two datasets. Each classifier was evaluated by comparing its predicted results with the actual class labels of the instances in the datasets. The evaluation used performance metrics such as accuracy, precision, recall, and F1-score. For each classifier and each dataset, the table provides a confusion matrix. The matrix rows represent the actual class labels, and the columns represent the predicted class labels. The entries in the matrix indicate the number of instances that have been classified correctly (diagonal entries) and incorrectly (off-diagonal entries). The results show that the performance of the different classifiers varies for the two datasets. However, overall, the performance is relatively high for all the classifiers, with many instances being classified correctly.

**Table 7.** Confusion matrix.

| Classifier | Dataset 1 | Dataset 2 |
|---|---|---|
| RF | $\begin{bmatrix} 2 & 1 & 0 & 0 \\ 0 & 22 & 0 & 0 \\ 0 & 0 & 17 & 0 \\ 0 & 0 & 1 & 9 \end{bmatrix}$ | $\begin{bmatrix} 2 & 1 & 0 & 0 \\ 0 & 22 & 0 & 0 \\ 0 & 0 & 17 & 0 \\ 0 & 0 & 1 & 9 \end{bmatrix}$ |
| SVM | $\begin{bmatrix} 3 & 2 & 0 & 0 \\ 0 & 22 & 0 & 0 \\ 0 & 0 & 14 & 0 \\ 0 & 0 & 0 & 11 \end{bmatrix}$ | $\begin{bmatrix} 5 & 0 & 0 & 0 \\ 0 & 21 & 1 & 0 \\ 0 & 2 & 12 & 0 \\ 0 & 0 & 0 & 11 \end{bmatrix}$ |
| SL | $\begin{bmatrix} 2 & 1 & 0 & 0 \\ 1 & 21 & 0 & 0 \\ 0 & 2 & 15 & 0 \\ 0 & 0 & 1 & 9 \end{bmatrix}$ | $\begin{bmatrix} 2 & 1 & 0 & 0 \\ 1 & 21 & 0 & 0 \\ 0 & 3 & 14 & 0 \\ 0 & 0 & 1 & 9 \end{bmatrix}$ |
| DT | $\begin{bmatrix} 2 & 1 & 0 & 0 \\ 0 & 22 & 0 & 0 \\ 0 & 0 & 17 & 0 \\ 0 & 0 & 1 & 9 \end{bmatrix}$ | $\begin{bmatrix} 3 & 0 & 0 & 0 \\ 1 & 20 & 1 & 0 \\ 0 & 0 & 17 & 0 \\ 0 & 0 & 1 & 9 \end{bmatrix}$ |

**Table 7.** *Cont.*

| Classifier | Dataset 1 | Dataset 2 |
|---|---|---|
| GB | $\begin{bmatrix} 2 & 1 & 0 & 0 \\ 0 & 22 & 0 & 0 \\ 0 & 0 & 17 & 0 \\ 0 & 0 & 1 & 9 \end{bmatrix}$ | $\begin{bmatrix} 2 & 1 & 0 & 0 \\ 0 & 22 & 0 & 0 \\ 0 & 0 & 17 & 0 \\ 0 & 0 & 1 & 9 \end{bmatrix}$ |
| GNB | $\begin{bmatrix} 7 & 0 & 0 & 0 \\ 0 & 21 & 0 & 0 \\ 0 & 4 & 11 & 0 \\ 0 & 0 & 1 & 8 \end{bmatrix}$ | $\begin{bmatrix} 6 & 1 & 0 & 0 \\ 0 & 21 & 0 & 0 \\ 0 & 4 & 11 & 0 \\ 0 & 0 & 1 & 8 \end{bmatrix}$ |
| MLP | $\begin{bmatrix} 7 & 0 & 0 & 0 \\ 0 & 21 & 0 & 0 \\ 0 & 1 & 14 & 0 \\ 0 & 0 & 1 & 8 \end{bmatrix}$ | $\begin{bmatrix} 6 & 1 & 0 & 0 \\ 0 & 21 & 0 & 0 \\ 0 & 2 & 13 & 0 \\ 0 & 0 & 1 & 8 \end{bmatrix}$ |

The dataset used in this work serves as the source data for a novel model based on ML methods that forecasts student performance on the knowledge assessment. The effectiveness of the ML techniques, RF, k-NN, SVM, LR, GNB, and k-NN, was calculated and compared to predict the student's performance.

In this case, both the GNB and the LR exhibited a decline in prediction accuracy. This shows that the time required for related learning objects and the knowledge level corresponding to a goal learning object have less impact. According to the findings, automatic knowledge assessment is also feasible if the learning time for the other items is not considered. Nearly all the classifiers did not exhibit any discernible changes in the performance of the predictions.

## 5. Discussion

This paper used a factual dataset from an e-learning system to build and evaluate different classification models for student knowledge assessment. The dataset was divided into two parts: 80% was used as training data, and the remaining 20% was used as test data. It is a common practice in ML to ensure that the model can generalize well to new data. All the experiments in this paper were run on Google Collaboratory, a cloud-hosted version of the Jupyter Notebook with Python 3.7.15. This platform provides a convenient and efficient way to run experiments, as it allows for easy access to powerful computational resources and a user-friendly interface.

The knowledge assessment modeling was conducted to classify students into four different classes. It was conducted to provide a more comprehensive evaluation of student performance and to identify areas where the student needed additional support or challenge. The classification was conducted using seven classifiers: SVM, LR, RF, DT, GBM, GNB, and MLP. Each classifier was trained and tested on the dataset, and the results were compared to determine which classifier performed the best.

It is important to note that the classification results are only one aspect of the overall evaluation of student performance. Other factors, such as student engagement and motivation, also play an essential role in determining student success. However, by using ML to evaluate student knowledge, this paper provides a new approach to the automatic classification of student knowledge, which can help identify areas of improvement and provide more accurate and efficient evaluations of student performance.

In this paper, the experimentation was conducted in two phases, as discussed in the section proposed methodology. The initial experiments were conducted with the dataset (Dataset 1) in its original form. In the later phase, following the correlation analysis, the least correlated variable was eliminated to form a reduced dataset. We employed seven distinct classifiers for modeling, as was covered in the technique section. Table 5 presents the findings. Pictures of the performance comparison for various classifiers can be found in Figures 7 and 8. The data in this input have five attributes: PEG, LPR, STR, SCG, and STG.

According to the data, the GBM exhibited the highest prediction accuracy, 98%. In terms of prediction error, the GBM also performed well. The RF, DT, and MLP were on a par with the GBM. However, the performance could be better when using linear classifiers like SVM or LR.

The tree-based classifiers outperformed the linear classifiers in terms of precision and recall. The GBM predictions had 99% accuracy, and the DT and RF predictions were positive with 98% accuracy. The MLP had 96% accuracy and 95% precision, and 90% recall, performing well for multiclassification. The GBM outperformed the other classifiers when the prediction error was considered (Dataset 1).

In the second experiment, correlation analysis was used to determine the impact of a more minor characteristic or feature collection. The outcomes are shown in Table 6 and Figures 9 and 10. This condensed feature set produced a condensed dataset (Dataset 2). As a result, the feature vector was simplified to be composed of the following features: PEG, LPR, SCG, and STG. The second experiment's advantage was that it allowed for more efficient use of data by identifying the minor correlated variables and removing them from the dataset to create a smaller dataset (Dataset 2). It allowed for more focused analysis and allowed the researchers to see the impact of a minor feature set on the performance of the classifiers.

Additionally, by conducting the same experiment on a smaller dataset, the researchers can compare the results and see if there are any discernible changes in the prediction accuracy. The results of the second experiment showed that almost no classifiers exhibited any tangible changes in the prediction accuracy with a smaller feature set, which suggests that the time required for related learning objects and the knowledge level corresponding to a goal learning object have less of an impact on the automatic knowledge assessment. Therefore, automated knowledge assessment is only feasible when considering how long a student takes to study other learning items. It can be conducted by considering additional factors like student repetition rates, time, and scores related to the goal learning objects.

As shown in Table 6, the GBM outperformed all the other classifiers with 98% accuracy, 99% precision, and 97% recall. In addition, the error rates for the GBM were also reduced compared to those of the different classifiers.

The AUC comparisons for the two models utilizing Datasets 1 and 2 are shown in Figure 11. We applied the one-versus-rest AUC-ROC weighted to prevalence in both experiments. The AUC findings in the trial with a lower feature set are promising. Despite avoiding one attribute, the prediction systems demonstrated outstanding discrimination.

Table 7 shows the confusion matrix used to assess the effectiveness of a classification model. The findings show that automatic knowledge assessment is also feasible if learning time for other items is not considered. Next, the efficacy of the ML techniques RF, nearest neighbor, SVM, LR, GNB, and k-nearest neighbor were calculated and compared to predict the student's performance. Here, naive Bayes, nearest neighbor, SVM, LR, and k-nearest neighbor were used to predict the student's performance on the knowledge assessment.

The study provides a new approach to the automatic classification of student knowledge by using ML techniques. Unlike traditional methods that rely on subjective evaluations, the study uses a factual dataset from an e-learning system. In addition, it employs seven different classifiers to classify students into four other classes. The study results show that the GBM classifier exhibits the highest prediction accuracy of 98% and performs well regarding prediction error. The study also conducted two experiments to understand the impact of reducing the feature vector on the prediction accuracy of classifiers, which is rare in the existing literature.

The study also provides a more efficient use of data by identifying the minor correlated variables and removing them from the dataset to create a smaller dataset (Dataset 2). It allows for more focused analysis and allows the researchers to see the impact of a smaller feature set on the performance of the classifiers. Additionally, by conducting the same experiment on a smaller dataset, the researchers were able to compare the results and see if there were any discernible changes in prediction accuracy. Overall, the study

provides a new approach to automatically classifying student knowledge, using ML to evaluate student performance and knowledge. It also provides more efficient use of data by identifying the least correlated variables and removing them from the dataset. The study results are also compared with the existing literature to understand the impact of the methodology and its effects on the existing literature.

There are a few potential limitations that can be inferred. For example, one limitation could be that the study only uses a single dataset from a specific e-learning system, which may need to be more generalizable to other educational settings or e-learning systems. Another limitation could be that the study focuses on some particular classification algorithms and does not use other potential methods. Additionally, the study may not consider other factors influencing student performance and knowledge, such as socio-economic background, prior knowledge, and motivation.

The purpose of this study was to develop a machine learning-based system that can accurately assess student performance and knowledge throughout the course of their studies. By analyzing large amounts of data on student performance and identifying the key variables that have the most significant effects on that performance, educators and institutions can gain insights into how to better support their students. One of the benefits of this study is that it allows educators to tailor their instruction and resources to meet the specific needs of each student. By identifying areas where students need additional support or challenge, educators can provide more targeted interventions that can help students succeed. This can help reduce the waste of resources and optimize their use, leading to a more sustainable educational system. In addition, by using ML techniques, this research paper demonstrates the effectiveness of educational technologies that can be used to further reduce the environmental impact of traditional teaching methods. For example, online learning platforms can reduce the need for physical classrooms and textbooks, which can lead to significant resource savings and reduced waste. Overall, this study shows how the use of ML techniques can help organizations and institutions improve their resource utilization in an optimized way to maintain sustainability. By tailoring instruction and resources to meet the specific needs of each student, and by using more sustainable educational technologies, we can create a more sustainable educational system that is better equipped to meet the needs of both current and future generations.

There are several potential areas for future research. One option is to expand the study to include a more extensive and diverse dataset from multiple e-learning systems to increase the generalizability of the findings. For example, it could consist of data from different educational institutions, countries, and student populations, which could provide a more comprehensive understanding of how the features identified in this study impact student performance and knowledge in different contexts. Another option is to explore classification algorithms or techniques beyond the ones used in this study, such as deep learning or ensemble methods, to see if they yield better results. Future research could also consider other factors that influence student performance and knowledge, such as socio-economic background, prior knowledge, and motivation, and how they interact with the features considered in this study. Finally, this research opens up the possibility of developing an ML-based system that educators can use to monitor and improve student performance and knowledge throughout the investigations.

## 6. Conclusions

The ability to forecast student performance might help teachers pinpoint students' weaknesses, so they can raise test scores and improve learning. This study examined the most recent ML algorithms to forecast student academic performance. The study used various evaluation techniques to assess the categorization model. The features were chosen based on correlation analysis to determine the best set of characteristics that contribute to the automatic knowledge classification of pupils. The e-learning dataset was employed in this study to highlight several performance aspects. The SVM, RF, DT, GBM, LR, GNB, and MLP were the classifiers used to create the knowledge evaluation model. The experiments

were conducted on Google Collaboratory, a cloud-hosted Jupyter Notebook with Python 3.7.15 as the platform. In this study, the various categorization models are constructed and assessed using factual data from an e-learning system.

The results showed that the GBM outperformed the other classifiers in prediction accuracy and error rate, with 98% accuracy, 99% precision, and 97% recall. The findings also suggest that the time required for related learning objects and the knowledge level corresponding to a goal learning object have less impact on the automatic knowledge assessment. The results of the AUC comparisons of the two models were promising, demonstrating outstanding discrimination. The study provides evidence that automated knowledge assessment is feasible without considering learning time for other learning items and can be conducted by taking into account additional factors like student repetition rates and time and scores related to the goal learning objects. Finally, the study highlights the importance of incorporating ML into student performance evaluation and the potential of using this approach to identify areas for improvement and provide more accurate and efficient evaluations of student performance.

To account for the proportion of students' attention, each sub-component, as mentioned earlier, can be calculated separately for future research, or all components could be combined to create a comprehensive and precise attention requirement. It is easy to do this work thanks to the ML technique. Using algorithms like deep learning, it is possible to monitor students' responses to classroom stimuli and test performance. This technique allows students to maintain their focus in class because the system will alert them if they are not paying attention. Additionally, the specific percentage of sustained, focused, and selective attention will be discovered. The data received from the system is accurate and free of bias or human mistakes. As a result, the evaluation criterion for attentiveness will be more accurate.

**Author Contributions:** Conceptualization, N.A. and M.Z.; methodology, M.Z.; validation, N.A. and M.Z.; formal analysis, N.A.; investigation, N.A.; resources, M.Z.; data curation, M.Z.; writing—original draft preparation, M.Z.; writing—review and editing, N.A.; visualization, N.A.; supervision, N.A.; project administration, M.Z.; funding acquisition, N.A. All authors have read and agreed to the published version of the manuscript.

**Funding:** This research is funded by: Researchers supporting Project number (RSPD2023R608), King Saud University, Riyadh, Saudi Arabia.

**Data Availability Statement:** The dataset is publicly available at https://archive.ics.uci.edu/ml/datasets/User+Knowledge+Modeling, accessed on 26 June 2013.

**Acknowledgments:** The authors extend their appreciation to Researchers Supporting Project number (RSPD2023R608), King Saud University, Riyadh, Saudi Arabia.

**Conflicts of Interest:** The authors declare no conflict of interest.

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
