# Peer review of "Evaluating Student Knowledge Assessment Using Machine Learning Techniques"

_sustainability, doi:10.3390/su15076229_

Round 1
Reviewer 1 Report (Previous Reviewer 3)
The authors did their best to deal with all the comments, and as a result, the paper improved considerably.
I have some more small comments and suggestions, but after accounting for these, the paper can in my opinion be considered for publication.
* line 11-12 is about student assessment, the following sentence (line 12-14) is about evaluating the assessment, and the following sentence (line 14-16) is explaining the importance of student assessment. Therefore, i would restructure the abstract somewhat: starting with assessment and its importance, then saying that therefore assessment methods should be evaluated, and then talking about the focus of this paper: an evaluation of an assessment procedure.
* line 27: i would drop part of the sentence, keeping: "In the second experiment, the least correlated variable is removed ..."
* line 35: "reduced feature set" instead of "minor feature set"
*line 42: i would drop "growing competition in education and the"
*line 141: i would write "This paper uses two experiments to understand ..." (i would not yet talk about the two datasets, because they are described only later; except for in the abstract, but the paper should be readable without reading the abstract)
*line 249: drop "other"
*line 250: "the distribution of students over classes"
*line 250: "Around 30% of students are in the class with high-performing students, 28% in the class with average-performing students, and 30% and 12% in classes with low and very low-performing students, respectively"
*table 2: the names of the two first columns should be switched
*line 269-275: to be dropped. Pearson's correlation does not have to be explained.
* line 377-382: this is still the model for a dichotomous dependent variable, whereas it was just mentiond that the dependent variable is a categorical with more than two categories. Moreover, note that you write "logic regression" instead of "logistic regression"
*Table 4: for the GNB approach, it is said that the 'default options were used. But what are these (what is the default option can depend on the software, and can change over time)
Author Response
Response Sheet
Evaluating Student Knowledge Assessment using Machine Learning Techniques
Response to Reviewers
Dear sir/madam
Thank you for giving us the opportunity to submit a revised draft of the manuscript “Evaluating Student Knowledge Assessment using Machine Learning Techniques”. We appreciate the time and effort that you and the reviewers dedicated to providing feedback on our manuscript and are grateful for the insightful comments on and valuable improvements to our paper. We have incorporated most of the suggestions made by the reviewers. Those changes are highlighted within the manuscript. Please see below, in blue, for a point-by-point response to the reviewers’ comments and concerns. All page numbers refer to the revised manuscript file with tracked changes.
Reviewer 1
The authors did their best to deal with all the comments, and as a result, the paper improved considerably. I have some more small comments and suggestions, but after accounting for these, the paper can in my opinion be considered for publication.
Author Response: Thank you very much for your feedback and for acknowledging the efforts we made to improve our paper based on the previous comments. We appreciate your time and constructive feedback. We will carefully consider any additional comments or suggestions you may have, and make any necessary revisions to the manuscript to address them. We are grateful for your support and encouragement, and we hope that the updated version of the paper will be suitable for publication. Thank you again for your valuable feedback and for considering our paper for publication.
* line 11-12 is about student assessment, the following sentence (line 12-14) is about evaluating the assessment, and the following sentence (line 14-16) is explaining the importance of student assessment. Therefore, i would restructure the abstract somewhat: starting with assessment and its importance, then saying that therefore assessment methods should be evaluated, and then talking about the focus of this paper: an evaluation of an assessment procedure.
Author Response: Thank you for pointing this out. We have now restructured the abstract’s starting sentences as per the comments by explaining about the importance of the student knowledge assessment and the importance of the evaluation of the student knowledge assessment. The updated abstract are as follows:
Abstract: The process of learning about a student's knowledge and comprehension of a particular subject is referred to as student knowledge assessment. It helps to identify areas where students need additional support or challenge and can be used to evaluate the effectiveness of instruction, make important decisions such as student placement and curriculum development, and monitor the quality of education. Evaluating student knowledge assessment is essential in measuring student progress, informing instruction, and providing feedback to improve student performance and enhance the overall teaching and learning experience. This research paper is designed to create a Machine Learning (ML)-based system that assesses student performance and knowledge throughout the course of the studies and pinpoints the key variables that have the most significant effects on that performance and expertise. Additionally, it describes the impact of running models with data that only contains key features on their performance. To classify the students, the paper employs seven different classifiers, including Support Vector Machines (SVM), Logistic Regression (LR), Random Forest (RF), Decision Tree (DT), Gradient Boosting Machine (GBM), Gaussian Naive Bayes (GNB) and Multi-Layer Perceptron (MLP). This paper carries out two experiments to see how best to replicate the automatic classification of student knowledge. In the first experiment, the dataset (Dataset 1) was used in its original state, including all five properties listed in the dataset, to evaluate performance indicators. In the second experiment, the correlation analysis is performed on dataset 1. The least correlated variable is re-moved from the dataset to create a smaller dataset (Dataset 2), and the same set of performance indicators is evaluated. Then, the performance indicators using dataset-1 and dataset-2 are com-pared. The GBM exhibits the highest prediction accuracy of 98%, according to dataset 1. In terms of prediction error, GBM also performs well. The accuracy of optimistic forecasts of student per-formance denoted as the performance indicator ‘precision’, is highest in GBM with 99% while DT, RF, and SVM are 98% accurate in their optimistic forecasts for dataset-1. The second experiment's findings demonstrated that practically no classifiers showed appreciable improvements in prediction accuracy with a minor feature set in dataset-2. It shows that the time required for related learning objects and the knowledge level corresponding to a goal learning object have less impact.
* line 27: i would drop part of the sentence, keeping: "In the second experiment, the least correlated variable is removed ..."
Author Response: Thank you for your feedback on my writing. I appreciate your suggestion to revise the sentence on line 27. I agree that it would be clearer to drop part of the sentence and keep only the essential information. We have modified the respective sentence as per the comment. That is:
In the second experiment, the least correlated variable is removed from the dataset to create a smaller dataset (Dataset 2), and the same set of performance indicators is evaluated.
* line 35: "reduced feature set" instead of "minor feature set"
Author Response: Thank you for pointing this out. We have changed the “minor feature set” to “reduced feature set” in the revised manuscript.
*line 42: i would drop "growing competition in education and the"
Author Response: Thank you for this suggestion. We have removed the “growing competition in education” from the respective sentence. The updated sentence is:
High-quality education requires both the educational system and students to meet high standards.
*line 141: i would write "This paper uses two experiments to understand ..." (i would not yet talk about the two datasets, because they are described only later; except for in the abstract, but the paper should be readable without reading the abstract).
Author Response: Thank you for your valuable feedback on our writing. I appreciate your suggestion to revise the sentence on line 141. Your suggestion to rephrase the sentence as "This paper uses two experiments to understand..." is a good one. It provides a clearer and more concise introduction to the main focus of the paper. I also agree with your point that it's better to introduce the two datasets later in the paper when they are described in more detail.
This paper uses the two experiments to understand the impact of reducing the feature vector on the prediction accuracy of classifiers and to identify the best technique for simulating students' automatic knowledge classification;
*line 249: drop "other"
Author Response: Thank you for this suggestion. We revised the respective sentence, that is:
Four classes of students were retrieved: Very Low, Low, Middle, and High.
*line 250: "the distribution of students over classes"
Author Response: Thank you for this suggestion. We have updated the respective sentence, that is:
The distribution of student over classes in the dataset is shown in Figure 3 below.
*line 250: "Around 30% of students are in the class with high-performing students, 28% in the class with average-performing students, and 30% and 12% in classes with low and very low-performing students, respectively"
Author Response: Thank you for suggesting this correction in the wording of the sentence. We have revised the particular sentence as per the comment.
*table 2: the names of the two first columns should be switched
Author Response: Thank you for pointing out this error. We have interchange the respective column labels.
*line 269-275: to be dropped. Pearson's correlation does not have to be explained.
Author Response: Thank you for your feedback on my writing. I appreciate your suggestion to remove the sentences on lines 269-275, as they explain Pearson's correlation, which is not necessary in this context. I agree with your point that Pearson's correlation is a well-known statistical concept and does not need to be explained in detail. Therefore, I have removed those sentences from my paper to improve its clarity and focus. Thank you for your helpful input.
* line 377-382: this is still the model for a dichotomous dependent variable, whereas it was just mentiond that the dependent variable is a categorical with more than two categories. Moreover, note that you write "logic regression" instead of "logistic regression"
Author Response: Thank you for bringing this to my attention. We apologize for the error in my writing. You are correct that we used the term "logic regression" instead of "logistic regression" in that section. We have corrected the respective error in the revised manuscript.
Also we have revised the respected explanation and equations to eliminate the description of dichotomous dependent variable to multiple variables. The updated text in the manuscript is:
The logistic function used in LR models is the softmax function, which converts the linear combination of independent variables into a probability distribution over the different categories. The softmax function is defined in equation 4:
(4)
where is the predicted probability of category , is the linear combination of independent variables for category , and is the total number of categories. To compute for each category , the LR model estimates the coefficients of the independent variables in a similar way to the binary case. However, instead of estimating a single set of coefficients, LR for multiple categories estimates a separate set of coefficients for each category, resulting in a matrix of coefficients.
LR for multiple categories models the probability of each category given a set of independent variables using the softmax function. The model estimates a matrix of coefficients to compute the linear combination of independent variables for each category, and standardizing data is necessary to avoid overfitting.
*Table 4: for the GNB approach, it is said that the 'default options were used. But what are these (what is the default option can depend on the software, and can change over time)
Author Response: Thank you for your comment and for bringing up this important point. We apologize for any confusion the statement may have caused. By "default options," we were referring to the default hyperparameters for the Gaussian Naive Bayes (GNB) classifier in the scikit-learn package, which was used in this study. we should have been more specific in describing these default hyperparameters.
For the GNB classifier, the default hyperparameters are: priors=None, var_smoothing=1e-9. We have updated the manuscript to reflect this information.

Reviewer 2 Report (Previous Reviewer 1)
This paper introduces e a Machine Learning (ML)-based system that assesses student performance and knowledge throughout the course of the studies. The manuscript's readability needs to be improved, so I have some questions after reading it.
The innovation point of the paper is not cohesive enough, and the structure is unclear. It is suggested to condense the innovation point of this paper.
The article lacks sufficient citations. It is recommended to include references to some new machine learning works: DOI: 10.1109TCYB.2022.3169773, DOI: 10.1109/TNNLS.2022.3149394.
Author Response
Response Sheet
Evaluating Student Knowledge Assessment using Machine Learning Techniques
Response to Reviewers
Dear sir/madam
Thank you for giving us the opportunity to submit a revised draft of the manuscript “Evaluating Student Knowledge Assessment using Machine Learning Techniques”. We appreciate the time and effort that you and the reviewers dedicated to providing feedback on our manuscript and are grateful for the insightful comments on and valuable improvements to our paper. We have incorporated most of the suggestions made by the reviewers. Those changes are highlighted within the manuscript. Please see below, in blue, for a point-by-point response to the reviewers’ comments and concerns. All page numbers refer to the revised manuscript file with tracked changes.
Reviewer 2
This paper introduces e a Machine Learning (ML)-based system that assesses student performance and knowledge throughout the course of the studies. The manuscript's readability needs to be improved, so I have some questions after reading it.
The innovation point of the paper is not cohesive enough, and the structure is unclear. It is suggested to condense the innovation point of this paper.
Author Response: Thank you for your feedback. I appreciate your suggestion that I condense the innovation point of my paper to make it more cohesive and clearer for the reader. To address this, I have revisit the organization of my paper and revised some portion of the manuscript to enhance the cohesiveness throughout the text. The added explanation is highlighted in the revised manuscript. Additionally, I have taken reviewer’s feedback into account and consider condensing the innovation points to improve the coherence of our writing. We understand that this will help to ensure that the reader can easily understand and follow the main ideas presented in the paper.
The article lacks sufficient citations. It is recommended to include references to some new machine learning works: DOI: 10.1109TCYB.2022.3169773, DOI: 10.1109/TNNLS.2022.3149394.
Author Response: Thank you for your feedback on my article. I appreciate your suggestion to include references to some new machine learning works. I will certainly take a look at the articles you mentioned, DOI: 10.1109TCYB.2022.3169773 and DOI: 10.1109/TNNLS.2022.3149394, and noticed that they are not fitting in the context of our work. I will consider the suggested articles in the future researches.
I understand the importance of providing sufficient citations to relevant literature to support the claims made in our article. We have already cited the necessary and appropriate references in the manuscript related to or in the context of our research topic.

This manuscript is a resubmission of an earlier submission. The following is a list of the peer review reports and author responses from that submission.
Round 1
Reviewer 1 Report
The manuscript presented to create a Machine Learning (ML)-based system that assesses student performance and knowledge throughout the course of the studies. The paper is well structured and concepts are clearly exposed. The overall fluency of the paper would benefit from more structured sentences. Nevertheless, the proposed paper needs some modifications to be at its best.
In Section 1 line 138-143, the motivation and contribution should be further summarized, including the advantages compared with traditional methods.
I suggest adding the advanced classifiers based on deep learning to the Introduction, which have been used for the multifield application, including feature extraction and classification. 10.1109/TGRS.2021.3097093, 10.1109/TNNLS.2022.3149394, 10.1109TGRS.2021.3093334, and 10.1109TCYB.2022.3169773.
In Section 6 line 496, in addition to the training set and testing set, whether the validation set is also used for model optimization.
I suggest supplementing the following parts of the discussion with proper citations: limitations of your study and recommendations for future research.
Author Response
Evaluating Student Knowledge Assessment using Machine Learning Techniques
Response to Reviewers
Dear sir/madam
Thank you for giving us the opportunity to submit a revised draft of the manuscript “Evaluating Student Knowledge Assessment using Machine Learning Techniques”. We appreciate the time and effort that you and the reviewers dedicated to providing feedback on our manuscript and are grateful for the insightful comments on and valuable improvements to our paper. We have incorporated most of the suggestions made by the reviewers. Those changes are highlighted within the manuscript. Please see below, in blue, for a point-by-point response to the reviewers’ comments and concerns. All page numbers refer to the revised manuscript file with tracked changes.
Reviewer 1
The manuscript presented to create a Machine Learning (ML)-based system that assesses student performance and knowledge throughout the course of the studies. The paper is well structured and concepts are clearly exposed. The overall fluency of the paper would benefit from more structured sentences. Nevertheless, the proposed paper needs some modifications to be at its best.
Comment 1: In Section 1 line 138-143, the motivation and contribution should be further summarized, including the advantages compared with traditional methods.
Author Response: Thank you for this suggestion. we have revised the contributions of this paper in the revised manuscript that is:
the major contribution of this paper is as follows:
- This paper aims to identify a group of characteristics or traits that influence students' automatic knowledge classification and also, to identify the role of ML in conceptualizing and evaluating student education, as well as the challenges and risks that need to be considered;
- This paper uses the two experiments to understand the impact of reducing the feature vector on the prediction accuracy of classifiers and to identify the best technique for simulating students' automatic knowledge classification;
- To identify a group of characteristics or traits that influence students' automatic knowledge classification, this study uses of seven different classifiers, including Support Vector Machines (SVM), Logistic Regression (LR), Random Forest (RF), Decision Tree (DT), Gradient Boosting Machine (GBM), Gaussian NB Classifier (GNB), and Multi-Layer Perceptron (MLP), providing a comprehensive evaluation of student knowledge;
- This paper presents the analysis for identifying the least correlated variables and re-moving them from the dataset to create a smaller dataset (Dataset 2) providing a more focused analysis and understanding of the impact of a smaller feature set on the performance of the classifiers;
Comment 2: I suggest adding the advanced classifiers based on deep learning to the Introduction, which have been used for the multifield application, including feature extraction and classification. 10.1109/TGRS.2021.3097093, 10.1109/TNNLS.2022.3149394, 10.1109TGRS.2021.3093334, and 10.1109TCYB.2022.3169773.
Author Response: Thank you for this suggestion. We respectfully disagree. We have already employed seven different classifiers, including Support Vector Machines (SVM), Logistic Regression (LR), Random Forest (RF), Gaussian NB Classifier (GNB), Decision Tree (DT), Gradient Boosting Machine (GBM) and Multi-Layer Perceptron (MLP) to classify and evaluate the student performance. It is very hard to used or implement further more classifier in this study. We will consider these suggested classifiers for the future researches.
Comment 3: In Section 6 line 496, in addition to the training set and testing set, whether the validation set is also used for model optimization.
Author Response: we appreciate the reviewer’s feedback. we have revised the section 6 (discussion). We have used only training and testing set in this study. The paper utilizes a factual dataset from an e-learning system, which is divided into two parts: 80% of the dataset is used as training data and the remaining 20% is used as test data. The research goal of this is to evaluate the performance of a specific classifier and identifying set of parameters, not to optimize it, so, the use of validation set would be inappropriate.
Comment 4: I suggest supplementing the following parts of the discussion with proper citations: limitations of your study and recommendations for future research.
Author Response: while we appreciate the reviewer’s feedback as well as the time and effort they give to review our paper. A paragraph for the limitation of this study has been added at the second last paragraph of the discussion section. That is:
There are a few potential limitations that can be inferred. One limitation could be that the study only uses a single dataset from a specific e-learning system, which may not be generalizable to other educational settings or e-learning systems. Another limitation could be that the study focuses on some specific classification algorithms and does not other more potential methods. Additionally, the study may not consider other factors that may influence student performance and knowledge, such as socio-economic background, prior knowledge, and motivation.
The recommendation for the future research is added at the end of the discussion section. That is:
There are several potential areas for future research. One option is to expand the study to include a larger and more diverse dataset from multiple e-learning systems to in-crease the generalizability of the findings. This could include data from different educational institutions, different countries, and different student populations, which could provide a more comprehensive understanding of how the features identified in this study impact student performance and knowledge in different contexts. Another option is to explore different classification algorithms or techniques beyond the ones used in this study, such as deep learning or ensemble methods, to see if they yield better results. Additionally, future research could also consider other factors that may influence student performance and knowledge, such as socio-economic background, prior knowledge, and motivation, and how they interact with the features considered in this study. Finally, this research opens up the possibility of developing a Machine Learning (ML) based system that can be used by educators to monitor and improve student performance and knowledge throughout the course of the studies.

Reviewer 2 Report
The article deals with a topic of interest, it is very well structured and I appreciate the list of references with the methodology used and results. The research methodology is clearly presented, as are the discussions and results. Proposals are made for future research.
Author Response
Evaluating Student Knowledge Assessment using Machine Learning Techniques
Response to Reviewers
Dear sir/madam
Thank you for giving us the opportunity to submit a revised draft of the manuscript “Evaluating Student Knowledge Assessment using Machine Learning Techniques”. We appreciate the time and effort that you and the reviewers dedicated to providing feedback on our manuscript and are grateful for the insightful comments on and valuable improvements to our paper. We have incorporated most of the suggestions made by the reviewers. Those changes are highlighted within the manuscript. Please see below, in blue, for a point-by-point response to the reviewers’ comments and concerns. All page numbers refer to the revised manuscript file with tracked changes.
Reviewer 2
Comment 1: The article deals with a topic of interest, it is very well structured and I appreciate the list of references with the methodology used and results. The research methodology is clearly presented, as are the discussions and results. Proposals are made for future research.
Author Response: We appreciate the reviewer’s feedback. We appreciate the time and effort that the reviewers dedicated to providing feedback on our manuscript.

Reviewer 3 Report
The authors created a machine learning system that assesses the performance of students. The paper describes the key variables that affect performance and knowledge, and compare the performance of different machine learning algorithms for predicting students’ performance.
Although the topic is interesting, I am rather doubtful about the value of the paper.
1. First, I am not convinced that this paper offers much valuable new insights.
2. Second, I found the text very hard to read, for several reasons.
a. One reason is that a lot of abbreviations are used in the paper. To make it even harder, several abbreviations are not introduced by writing the term in full the first time (e.g., DT, MLP, LMS, AIML, k-NN, …). In addition, for some abbreviations there is a description but it is not immediately clear what the acronym stands for (e.g., line 235: “The exam scores that students received for study materials associated with goal objects and specifically for goal items, respectively, were LPR and PEG.”. Several times, I had to look back in the text to refresh my mind about the meaning of abbreviations.
b. A second reason is that the train of thought is not always clear. To give an example, in the abstract it is suggested that the following approaches are compared: SVM, LR, RF, and ANN (so GBM is not listed as one of the techniques that are compared), but two sentences later it is said that GBM seems to outperforms other approaches. As another example: Figure 1 and Figure 2 (and Figure 5) look somewhat similar. I would further clarify what the differences and links are between these schemes. A third example: in their second section, the authors describe the results of some studies, and mention for some studies that “they did not adhere to the requirements for a systematic literature analysis”. It is not clear why this remark is made; it made me think that the authors wanted to give an overview of review studies only, but also other studies (Taglietti et al., 2021) are mentioned that are not review studies. These are only a few examples of how the reader has to try to reconstruct what has been done or what the authors mean (both for understanding individual sentences and for understanding the red line through the paper).
c. Third, the paper does not give a clear description of the many techniques that are used. For instance, line 92-93 says: “The fundamental concept of this technique [GBM] is to build the new base-learners to have a maximum correlation with the ensemble's overall negative gradient of the loss function.” Such a sentence can not be understood unless you are already an expert in the field. I understand that it is not possible (and not desirable) to introduce all techniques in detail, but the authors could try to clarify the basic principles and for further details refer to other sources.
d. Finally, some sentences are hard to understand, sound weird or seem to be incomplete. As an example, it is not clear to me how to interpret the last sentence of the abstract: “The same set of evaluation metrics yields results.”
Other remarks (some of them are detailed remarks, but these illustrate again the lack of clarity):
· * abstract: The authors argue that there is a distinction between assessment and evaluation. While this is correct, the explanation is unclear and this distinction is not further elaborated in the text, so I think there is no need to explain that distinction in the abstract. Lines 10-16 can in my opinion simply be dropped.
· * line 219: it is not clear what the role is of Table 1: what studies are included, and why. Moreover, the information is sometimes hard to understand. For instance, in the column ‘Results’, I see ‘Creating prediction models’ or ‘A semi-AdaBoost technique is adopted. On 12 UCI machine learning datasets.’ These are not research results, but look rather like methods.
· * line 221: “In this study, machine learning classification models are developed and evaluated using the user knowledge modelling data set for an e-Learning environment [34]. [34] gathers the data using a six-step process, updating and evaluating the pupils' knowledge at each stage [35].” In this section, the authors seem to describe their study, but at the same time they refer to what is done in other studies. It is not clear what the role is of the description of what was done in other studies: does this mean that the same was done in this study? The same remark applies to a few other sentences in this section.
· * In general it is hard to follow what the input and the output of the techniques is. For instance, it is not clear (to me) where the four different student types (line 230) come from: are these the results of the k-NN approach, or are these measured directly? And what is the relation between these ‘user knowledge classes’ and the variable UNS that refers to the ‘user’s knowledge level’? These unclarities makes it also hard to understand what the purpose is of the different analyses presented, and more general the purpose of the paper.
· * In the description of the classifiers, some techniques are described for dichotomous dependent variables (e.g., the logistic regression, line 327, or the confusion matrix in line 376) whereas the membership for 4 classes is predicted. In general, linking more explicitly the techniques to this context would help the reader (e.g., for the decision tree approach, it could be clarified what the Leaf nodes are and what a decision node could look like in this application).
· * Line 417: it is not completely clear what the rationale is to add a second stage to the analysis.
· * Figure 7 (line 434): GB should be GBM
· * Line 446: “The correlation analysis between the dataset's 446 properties is shown in Figure 8.” Figure 8 does not present correlations.
· * Figure 9: “Performance” instead of “Performance”
· * The Discussion-section is not really a discussion section, but describes (again) the results, referring to the tables and figures. Similarly, the Conclusions-section does not simply draw a final conclusion, but includes some elements that should be in the Results’ section or to the Discussion-section.
Author Response
Evaluating Student Knowledge Assessment using Machine Learning Techniques
Response to Reviewers
Dear sir/madam
Thank you for giving us the opportunity to submit a revised draft of the manuscript “Evaluating Student Knowledge Assessment using Machine Learning Techniques”. We appreciate the time and effort that you and the reviewers dedicated to providing feedback on our manuscript and are grateful for the insightful comments on and valuable improvements to our paper. We have incorporated most of the suggestions made by the reviewers. Those changes are highlighted within the manuscript. Please see below, in blue, for a point-by-point response to the reviewers’ comments and concerns. All page numbers refer to the revised manuscript file with tracked changes.
Reviewer 3
The authors created a machine learning system that assesses the performance of students. The paper describes the key variables that affect performance and knowledge, and compare the performance of different machine learning algorithms for predicting students’ performance.
Although the topic is interesting, I am rather doubtful about the value of the paper.
Comment 1: First, I am not convinced that this paper offers much valuable new insights.
Author Response: The paper provides valuable new insights as compared to the existing literature by using machine learning techniques to evaluate student performance and knowledge. The paper utilizes a factual dataset from an e-learning system, which is divided into two parts: 80% of the dataset is used as training data and the remaining 20% is used as test data. The knowledge assessment modeling is done as a classification of students into four different classes. To classify the students, the paper employs seven different classifiers, including Support Vector Machines (SVM), Logistic Regression (LR), Random Forest (RF), Gaussian NB Classifier (GNB), Decision Tree (DT), Gradient Boosting Machine (GBM) and Multi-Layer Perceptron (MLP). The paper also provides valuable insights by conducting two experiments to understand the impact of reducing the feature vector on the prediction accuracy of classifiers, which is not common in the existing literature. The paper also provides insights by identifying the least correlated variables and removing them from the dataset to create a smaller dataset (Dataset 2). The results of the study are also compared with the existing literature to understand the impact of the methodology and results on the existing literature.
Comment 2: Second, I found the text very hard to read, for several reasons.
- One reason is that a lot of abbreviations are used in the paper. To make it even harder, several abbreviations are not introduced by writing the term in full the first time (e.g., DT, MLP, LMS, AIML, k-NN, …). In addition, for some abbreviations there is a description but it is not immediately clear what the acronym stands for (e.g., line 235: “The exam scores that students received for study materials associated with goal objects and specifically for goal items, respectively, were LPR and PEG.”. Several times, I had to look back in the text to refresh my mind about the meaning of abbreviations
Author Response: Thank you for this feedback. we have revised the abbreviations in the manuscript and expressed them with its full form when they first appear.
Learning Management System (LMS),
Educational Data Mining (EDM),
Artificial Intelligence & Machine Learning (AIML),
Systematic Literature Review (SLR),
Reliability Standard (RS),
Support Vector Machine (SVM),
Information and Communication Technologies (ICT),
undergraduate program at the Federal University of Santa Catarina (UFSC),
k-nearest neighbour (k-NN),
artificial neural network (ANN)
In this research, we have used the same attributes as the ones commonly used in the existing literature. The attributes such as STG, SCG, STR, LPR, PEG, and UNS are commonly used in the literature to evaluate student knowledge and performance, so we have employed these attributes in the same way as they have been used in previous studies. This allows for consistency and comparability with previous research in the field.
- A second reason is that the train of thought is not always clear. To give an example, in the abstract it is suggested that the following approaches are compared: SVM, LR, RF, and ANN (so GBM is not listed as one of the techniques that are compared), but two sentences later it is said that GBM seems to outperforms other approaches. As another example: Figure 1 and Figure 2 (and Figure 5) look somewhat similar. I would further clarify what the differences and links are between these schemes. A third example: in their second section, the authors describe the results of some studies, and mention for some studies that “they did not adhere to the requirements for a systematic literature analysis”. It is not clear why this remark is made; it made me think that the authors wanted to give an overview of review studies only, but also other studies (Taglietti et al., 2021) are mentioned that are not review studies. These are only a few examples of how the reader has to try to reconstruct what has been done or what the authors mean (both for understanding individual sentences and for understanding the red line through the paper).
Author Response: We appreciate the reviewer’s feedback.
The reviewer is right. We did not include the GBM as an employed classifier previously. We have now rewritten the Abstract section and fixed such inaccuracies. We have now explicitly described the classifiers that were employed.
Figure 1 depicts an overview of the study strategy connected to assessing student understanding. Figure 2 depicts the machine learning processes used in evaluating student performance assessment. Figure 5 also depicts the flowchart of the recommended technique for this research project. In contrast to the theme of this paper, these figures describe several phases and procedures.
In the third example of the comment, reviewer is right. The information not completely provided. Now we have added more explanation for this in the revised manuscript. that is:
“However, the authors of this study did not adhere to the standards for a systematic literature review, which typically involves a rigorous and systematic search, appraisal, and synthesis of the literature on a topic. Additionally, the study did not focus specifically on Machine Learning (ML) methods, which is a specific field of study within EDM that involves the use of ML techniques to analyze educational data. This means that the study did not provide a comprehensive and in-depth examination of the use of ML methods in the field of EDM.”
Comment 3: Third, the paper does not give a clear description of the many techniques that are used. For instance, line 92-93 says: “The fundamental concept of this technique [GBM] is to build the new base-learners to have a maximum correlation with the ensemble's overall negative gradient of the loss function.” Such a sentence cannot be understood unless you are already an expert in the field. I understand that it is not possible (and not desirable) to introduce all techniques in detail, but the authors could try to clarify the basic principles and for further details refer to other sources.
Author Response: Thank you for this suggestion. we have added more explanation for the techniques used in this study in order to clarify the basic principle and understanding of the particular technique. The added explanation is highlighted in the revised manuscript. that is:
- Logistic Regression (LR)
Logistic Regression (LR) is a statistical method used for solving classification prob-lems. It is a type of generalized linear model that is used to model the relationship be-tween a binary outcome variable and a set of input features. The basic principle of LR is to find the best linear combination of the input features that maximizes the likelihood of the observed outcome.
In the context of Evaluating Student Knowledge Assessment, Logistic Regression (LR) can be used to predict students' proficiency levels in a particular learning object. It can al-so be used to predict the students' performance in future assessments, which can be used to improve the teaching and learning process. LR is a simple, interpretable, and efficient model that can handle both linear and non-linear relationships between the student's performance and knowledge and the characteristics that influence them. Additionally, LR can be used for multi-class problems, in which the dataset has more than two classes.
- Decision Tree Classifier (DT)
A Decision Tree Classifier (DT) is a type of algorithm used for supervised learning tasks such as classification and regression. It works by recursively partitioning the dataset into subsets based on the values of the input features. The goal is to create partitions that will result in the greatest separation of the classes. The partitioning process results in a tree-like structure, with each internal node representing a feature test and each leaf node representing a class label. The basic principle of DT is that it learns to approximate any complex function by training on a set of input-output pairs. The training process is based on finding the best feature test at each internal node that will result in the greatest separation of the classes. The final decision tree is a set of feature tests, with each test representing a path from the root of the tree to a leaf node.
In the context of Evaluating Student Knowledge Assessment, Decision Tree Classifier (DT) can be used to classify students into different levels of proficiency in a particular learning object, such as novice, beginner, intermediate or expert. DT can also be used to predict the student's performance in future assessments, which can be used to improve the teaching and learning process. DT can learn non-linear relationships between the student's performance and knowledge, and the characteristics that influence them, and it is simple to interpret and explain the results. Additionally, DT can be used to classify multiclass problems, in which the dataset has more than two classes.
- Gradient Boosting Machine (GBM)
Gradient Boosting Machine (GBM) is a ML algorithm that is used for supervised learning tasks, such as classification and regression. It is a type of ensemble learning method, which combines the predictions of multiple base models to improve the overall performance. GBM works by iteratively adding new base models, such as decision trees, to the ensemble, where each new model is trained to correct the errors made by the previous models. The algorithm uses gradient descent to minimize the errors made by the ensemble. In each iteration, the algorithm fits a new model to the negative gradient of the loss function, which is a measure of the error made by the ensemble. The final ensemble is a weighted sum of the base models, where the weights are determined by the training process.
In the context of Evaluating Student Knowledge Assessment, GBM can be used to classify students into different levels of proficiency in a particular learning object, such as novice, beginner, intermediate or expert. By using GBM, the system can automatically learn the relationship between the student's performance and knowledge, and the charac-teristics that influence them. GBM can also be used to predict the student's performance in future assessments, which can be used to improve the teaching and learning process.
- Multi-layer Perceptron (MLP)
A MLP is a type of artificial neural network (ANN) that is used for supervised learning tasks such as classification and regression. It is composed of an input layer, one or more hidden layers, and an output layer. Each layer is made up of a set of neurons, which are connected to the neurons in the previous and next layers via a set of weights.
The basic principle of MLP is that it learns to approximate any complex function by training on a set of input-output pairs. The training process is based on adjusting the weights of the connections between the neurons in order to minimize the error between the predicted output and the true output. In the context of Evaluating Student Knowledge Assessment, MLP can be used to classify students into different levels of proficiency in a particular learning object, such as novice, beginner, intermediate or expert. MLP can also be used to predict the student's performance in future assessments, which can be used to improve the teaching and learning process. MLP can learn non-linear relationships be-tween the student's performance and knowledge, and the characteristics that influence them. Additionally, MLP can be used to classify multi-class problems, in which the dataset has more than two classes.
Comment 4: Finally, some sentences are hard to understand, sound weird or seem to be incomplete. As an example, it is not clear to me how to interpret the last sentence of the abstract: “The same set of evaluation metrics yields results.”
Author Response: Thank you for pointing this out. The reviewer is correct; we have corrected the sentences of this manuscript at our best.
“Results of this experimentation are obtained using the same set of assessment parameters such as accuracy, precision, and recall.”
Comment 5: Other remarks (some of them are detailed remarks, but these illustrate again the lack of clarity):
- abstract: The authors argue that there is a distinction between assessment and evaluation. While this is correct, the explanation is unclear and this distinction is not further elaborated in the text, so I think there is no need to explain that distinction in the abstract. Lines 10-16 can in my opinion simply be dropped.
Author Response: Thank you for this suggestion. we agree with the reviewer that there is no need to explain that distinction in the abstract. So, we have removed the distinction between assessment and evaluation related phrases from the abstract and added other introductory phrases in the abstract section. The updated abstract is:
Abstract: The process of learning about a student's knowledge and comprehension of a particular subject is referred as student knowledge assessment. Evaluating student knowledge assessment is essential in measuring student progress, informing instruction, and providing feedback to improve student performance and enhance the overall teaching and learning experience. It helps to identify areas where students need additional support or challenge and can be used to evaluate the effectiveness of instruction, make important decisions such as student placement and curriculum development, and monitor the quality of education. This research paper is designed to create a Machine Learning (ML)-based system that assesses student performance and knowledge throughout the course of the studies and pinpoints the key variables that have the biggest effects on that performance and knowledge. Additionally, it describes the impact of running models with data that only contains key features on their performance. To classify the students, the pa-per employs seven different classifiers, including Support Vector Machines (SVM), Logistic Regression (LR), Random Forest (RF), Decision Tree (DT), Gradient Boosting Machine (GBM), Gaussian NB Classifier (GNB) and Multi-Layer Perceptron (MLP). This paper carries out an experiment to see how best to replicate the automatic classification of student knowledge. In this situation, GBM exhibits the greatest 98% prediction accuracy. In terms of prediction error, GBM also performs well. In terms of performance, RF, DT, and MLP are on par with GBM. The performance is relatively poor when using linear classifiers like SVM or LR. The three-based classifier, which is superior than linear classifiers, will thereafter be the focus of our effort. While DT and GBM are 98% accurate in their optimistic forecasts, RF is 98% accurate. With this dataset, MLP with 96% accuracy, 95% precision, and 90% recall also performs well. The Gradient Booster outperforms other classifiers when the prediction error is taken into account. We will also dis-cuss the least correlated feature. The feature vector is reduced as a smaller dataset. The reduced dataset is processed using the same methods as the initial experiment. Results of this experimentation are obtained using the same set of assessment parameters such as accuracy, precision, and recall.
Comment 6: line 219: it is not clear what the role is of Table 1: what studies are included, and why. Moreover, the information is sometimes hard to understand. For instance, in the column ‘Results’, I see ‘Creating prediction models’ or ‘A semi-AdaBoost technique is adopted. On 12 UCI machine learning datasets.’ These are not research results, but look rather like methods.
Author Response: Thank you for pointing this out. we regret this type of mistakes and we have updated the table 1 as per the comments. The updated or added texts are highlighted in the revised manuscript.
Comment 7: line 221: “In this study, machine learning classification models are developed and evaluated using the user knowledge modelling data set for an e-Learning environment [34]. [34] gathers the data using a six-step process, updating and evaluating the pupils' knowledge at each stage [35].” In this section, the authors seem to describe their study, but at the same time they refer to what is done in other studies. It is not clear what the role is of the description of what was done in other studies: does this mean that the same was done in this study? The same remark applies to a few other sentences in this section.
Author Response: Thank you for pointing this out. The reviewer is correct, this sentence creating confusion. Actually, this happens due to the wrong referencing. Now, we have corrected the sentences in the revised manuscript. that is:
“In this study, machine learning classification models are developed and evaluated using a dataset for an e-Learning environment. The dataset was gathered using a six-step process, updating and evaluating the pupils' knowledge at each stage. The dataset shows five different student characteristics that were noticed when they interacted with the learning environment.”
Comment 8: In general it is hard to follow what the input and the output of the techniques is. For instance, it is not clear (to me) where the four different student types (line 230) come from: are these the results of the k-NN approach, or are these measured directly? And what is the relation between these ‘user knowledge classes’ and the variable UNS that refers to the ‘user’s knowledge level’? These unclarities makes it also hard to understand what the purpose is of the different analyses presented, and more general the purpose of the paper.
Author Response: Thank you for pointing this out. The reviewer is correct, this confusion is due to the wrong referencing and presentation of sentences. The four different student classes are retrieved from the datasets using k-NN approach. We have done some rearrangement of the paragraphs in the revised manuscript to remove these type of unclarities.
“In this work, the user knowledge classes were classified using the k-nearest neighbor approach, a hybrid machine learning methodology that combines k-NN with meta-heuristic exploration approaches. Four different classes of students were retrieved, that is: Very Low, Low, Middle, and High.”
Comment 9: In the description of the classifiers, some techniques are described for dichotomous dependent variables (e.g., the logistic regression, line 327, or the confusion matrix in line 376) whereas the membership for 4 classes is predicted. In general, linking more explicitly the techniques to this context would help the reader (e.g., for the decision tree approach, it could be clarified what the Leaf nodes are and what a decision node could look like in this application).
Author Response: Thank you for this suggestion. we have added some explanations for different techniques. In particular, we have added the text for the decision tree approach, that is:
Here, the decision node represents “Is student performance greater than a certain threshold?" and the branches from that node would lead to different leaf nodes representing different classifications of student knowledge. The decision tree classifier uses these decision nodes to iteratively split the data into subsets, with each split becoming more specific and leading to a more accurate prediction of student knowledge.
Comment 10: Line 417: it is not completely clear what the rationale is to add a second stage to the analysis.
Author Response: We appreciate the reviewer’s feedback. We regret this type of error, actually, this is happing due to the wrong placements of figures in the manuscript. We have rearranged the result section in order to eliminate these type of unclarities.
The advantage of performing the second experiment is that it allows for a more efficient use of data by identifying the least correlated variables and removing them from the dataset to create a smaller dataset (Dataset 2). This allows for a more focused analysis and allows the researchers to see the impact of a smaller feature set on the performance of the classifiers. Additionally, by conducting the same experiment on the smaller dataset, the researchers can compare the results and see if there are any discernible changes in prediction accuracy. The results of the second experiment showed that nearly no classifiers exhibit any discernible changes in prediction accuracy with a smaller feature set, which suggests that the time required for related learning objects and the knowledge level corresponding to a goal learning object have less of an impact on the automatic knowledge assessment. This means that automatic knowledge assessment is still feasible without considering how long a student took to study other learning items, and can be done by taking into account additional factors like student repetition rates and time and score related to the goal learning objects.
Comment 11: Figure 7 (line 434): GB should be GBM
Author Response: Thank you for pointing this out. We have corrected the respective error in the manuscript.
Comment 12: Line 446: “The correlation analysis between the dataset's 446 properties is shown in Figure 8.” Figure 8 does not present correlations.
Author Response: Thank you for pointing this out. there was a wrong in-text-citation. Now we have corrected this in-text-citation of figure 8 in the revised manuscript.
“The error measure comparison of classifiers is shown in Figure 8.”
Comment 13: Figure 9: “Performance” instead of “Performance”
Author Response: Thank you for pointing this out. The reviewer is correct; we have corrected this spelling mistake in the revised manuscript.
Comment 14: The Discussion-section is not really a discussion section, but describes (again) the results, referring to the tables and figures. Similarly, the Conclusions-section does not simply draw a final conclusion, but includes some elements that should be in the Results’ section or to the Discussion-section.
Author Response: Thank you for this suggestion. We have added more explanation in the discussion section. Added text are given below.
In this paper, a factual dataset from an e-learning system is used to build and evaluate different classification models for the purpose of student knowledge assessment. The dataset is divided into two parts: 80% of the dataset is used as training data, and the remaining 20% is used as test data. This is a common practice in machine learning to ensure that the model is able to generalize well to new data. All of the experiments in this paper are run on Google Col-laboratory, a cloud-hosted version of Jupyter Notebook with Python 3.7.15. This platform provides a convenient and efficient way to run the experiments, as it allows for easy access to powerful computational resources and a us-er-friendly interface.
The knowledge assessment modeling is done as a classification of students into four different classes. This is done to provide a more comprehensive evaluation of student performance and to identify areas where the student needs additional support or challenge. The classification is done by using 7 classifiers, which include Support Vector Machines (SVM), Logistic Regression (LR), Random Forest (RF), Gaussian NB Classifier (GNB), Decision Tree (DT), Gradient Boosting Machine (GBM) and Multi-layer Perceptron (MLP). Each classifier is trained and tested on the dataset and the results are compared to deter-mine which classifier performs the best.
It is important to note that the results of the classification are only one aspect of the overall evaluation of student performance, and other factors such as student engagement and motivation also play an important role in determining student success. However, by using machine learning to evaluate student knowledge, this paper provides a new approach to the automatic classification of student knowledge, which can be useful in identifying areas of improvement and providing more accurate and efficient evaluations of student performance.
The advantage of performing the second experiment is that it allows for a more efficient use of data by identifying the least correlated variables and removing them from the dataset to create a smaller dataset (Dataset 2). This allows for a more focused analysis and allows the researchers to see the impact of a smaller feature set on the performance of the classifiers. Additionally, by conducting the same experiment on the smaller dataset, the researchers can compare the results and see if there are any discernible changes in prediction accuracy. The results of the second experiment showed that nearly no classifiers exhibit any discernible changes in prediction accuracy with a smaller feature set, which suggests that the time required for related learning objects and the knowledge level corresponding to a goal learning object have less of an impact on the automatic knowledge assessment. This means that automatic knowledge assessment is still feasible without considering how long a student took to study other learning items, and can be done by taking into account additional factors like student repetition rates and time and score related to the goal learning objects.
The study provides a new approach to the automatic classification of student knowledge by using machine learning techniques. Unlike traditional methods that rely on subjective evaluations, the study uses a factual dataset from an e-learning system and employs seven different classifiers to classify students into four different classes. The results of the study show that the GBM classifier exhibits the highest prediction accuracy of 98% and performs well in terms of prediction error. The study also conducts two experiments to understand the impact of reducing the feature vector on the prediction accuracy of classifiers, which is not common in the existing literature.
The study also provides a more efficient use of data by identifying the least correlated variables and removing them from the dataset to create a smaller dataset (Dataset 2). This allows for a more focused analysis and allows the researchers to see the impact of a smaller feature set on the performance of the classifiers. Additionally, by conducting the same experiment on the smaller dataset, the researchers can compare the results and see if there are any discernible changes in prediction accuracy. Overall, the study provides a new approach to the automatic classification of student knowledge and it uses machine learning to evaluate student performance and knowledge. It also provides a more efficient use of data by identifying the least correlated variables and removing them from the dataset. The results of the study are also compared with the existing literature to understand the impact of the methodology and results on the existing literature.

Round 2
Reviewer 3 Report
The authors revised the paper, and partly accounted for the comments I gave in the previous round. The text definitely has improved, but I am nevertheless disappointed.
There are still a lot of unclarities, and the authors were too sloppy in writing the paper, and again in revising the paper. Below, I will give other examples than the examples I gave before. I was frustrated to see that the authors made some changes based on the examples I gave before but failed to correct other errors and unclarities. The reader still has to guess what the authors mean at several places, or have to derive this from information that is given elsewhere. I will give other examples below, but again these are only examples. I do not consider it is as the responsibility of a reviewer to list all errors and to list all unclear sentences, so I rather expect that the authors really go through the whole manuscript again to make sure that everything is correct and clear. I also propose that they ask one or two persons who were not involved in this study to read the paper in order to evaluate the readability of all parts and sentences. So here again a few examples:
· * in the abstract (line 38) they write “While DT and GBM are 98% accurate in their optimistic forecasts, RF is 98% accurate.” The word “while” suggests that there will be a contrast, that DT and GBM are performing differently compared to RF, yet the three percentages are the same. Moreover, I tried to find these values in Tables 5 and 6, but I could not find these values there.
· * line 262-263: “were classified using the k-nearest neighbour (k-NN) approach, a hybrid machine learning methodology that combines k-NN with meta-heuristic exploration approaches.” This is unclear and not logical: k-NN is a combination of k-NN and other approaches?
· * line 267: “Around 30% of students are in classes with a high distribution,”. This is unclear: what is meant by a high distribution? And in fact all 30% of the students belong to one class only. I think the authors mean that 30% of the students are in the class with high-performing students.
· * line 289: “The Pearson correlation methodology, which assigns a value between 0 and 1, with 1 signifying total positive correlation and 0 denoting total negative correlation”. Pearson correlations go from -1 to +1, with -1 refer to a perfectly negative linear association and 0 referring to no linear association.
· * in the results-section first the data are described for the dataset with 5 attributes, and then for the reduced dataset with 4 attributes. Yet, Table 5 and 6 (for the first and second dataset, respectively) both mention 7 attributes in the title. Idem for Figure 9 and 11.
One of my suggestions was to make sure that abbreviations are written in full the first time, and afterwards the abbreviation is used consistently. Yet, when I look for instance at the term “Machine Learning”, I see that the first time it is used, the following is written (like it should be): Machine learning (ML). Subsequently, often the abbreviation is used (e.g., line 57, 58, 60), but at many other places machine learning is still written in full (line 67, 98, 103, 119, …). On line 715, again Machine learning (ML) is written. So I would like to ask the authors to use the search-function in their program, to make sure that these inconsistencies are found and removed! The same remark is true for other abbreviations (for instance the Gradient booster machine is sometimes but not systematically written as GBM and sometimes Gradient Booster is written).
Still the authors did not say what the abbreviations of the attribute names (STG, SCG, STR, LPR, PEG and UNS) stand for. The authors say that these abbreviations are commonly used, but I guess we can not assume that the readers of this journal know these abbreviations.
Another comment I had is that the column “Results” of the table with an overview of previous studies does not always give results. The authors changed the example I gave, but did not revise the other cells in that column! For instance: “Analysed the 80 significant research on predicting student performance in this survey.” is not a result of the study, but rather something that was done in that study.
Also my remark that the description of some techniques focuses on binary data remains the same. Still logistic regression is described for a dichotomous dependent variable (adding at the end the sentence that it can be used for multi-class problems, does not clarify much). Also the confusing matrix that is described is still for dichotomous outcomes.
The section “3. Dataset” is not only about the dataset, but is about the methods used, so should receive the title “Methods”. Moreover, I would start that section by describing in a few sentences the general strategy used. I think that the strategy was that based on a few knowledge variables, students were divided in four classes reflecting their level of performance, and that next 7 machine learning techniques were used to try to recover as good as possible the distribution of the students across the four classes, using five student attributes.
Details:
· * line 37: tree instead of three
· * line 49-50: I do not see how an increased competition calls for raising the standards. On the contrary, by raising the standards, there will be even more competition.
· * line 154: “the paper uses the two experiments”. Which two experiments? This is not clear from the preceding text.
· * line 177: “the second section” instead of “then the section”
· * line 255: “The dataset was gathered using a six-step process”. This is not clear to me. What are these steps?
· * line 349: “The highest is taken into account …”. The highest what?